

# Deep Transfer Learning Method for Seasonal TROPOMI XCH4 Albedo Correction

Alexander C. Bradley[1, 2]; Barbara Dix[2]; Fergus Mackenzie[3]; J. Pepijn Veefkind[4, 5]; and Joost A. de Gouw[1,2]

[1]Cooperative Institute for Research in Environmental Sciences, University of Colorado, Boulder, CO, 80309
[2]Chemistry Department, University of Colorado, Boulder, CO, 80309
[3]BlueSky Resources, Boulder CO, 80302, USA
[4]Royal Netherlands Meteorological Institute (KNMI), De Bilt, The Netherlands
[5]Delft University of Technology, Delft, The Netherlands

*Correspondence to:* Joost de Gouw (Joost.deGouw@colorado.edu)

**Abstract.** The retrieval of methane from satellite measurements is sensitive to the reflectance of the surface. In many regions, especially those with agriculture, surface reflectance depends on season, but this is not accounted for in many satellite products. It is an important issue to consider, as agricultural emissions of methane are significant and other sources, like oil and gas production, are also often located in agricultural lands. In this work, we use a set of 12 monthly machine learning models to

generate a seasonally resolved surface albedo correction for TROPOMI methane data across the Denver-Julesburg basin. We found that land cover is important in the correction, specifically the type of crops grown in an area, with drought-resistant crop covered areas requiring a correction of 5-6 ppb larger than areas covered in water-intensive crops. Additionally, the correction over different land covers changes significantly over the seasonally resolved timescale, with corrections over drought-resistant crops being up to 10 ppb larger in the summer than in the winter. This correction will allow for more accurate determination

of methane emissions by removing the effect of agricultural and other seasonal effects on the albedo correction. The correction may also allow for the deconvolution of agricultural methane emissions, which are seasonally dependent, from oil and gas emissions, which are more constant in time.

## 1 Introduction

The second most significant anthropogenic greenhouse gas (GHG), methane, has important climate implications.

Providing 27 times the warming potential of carbon dioxide on a 100-year timescale, but with a much shorter lifetime of less than 10 years, a reduction in methane emissions could ease global warming and potentially help achieve 1.5 or 2 degree goals (Boucher et al., 2009; Collins et al., 2018). Agriculture is the largest contributor to global anthropogenic methane emissions (50.63%), followed by the energy sector (28.65%) (Karakurt et al., 2012). Methane emissions from the energy sector are dominated by oil and gas operations which, in the United States, are still expanding following phase-out trends in coal-fired

power production (U.S. Energy Information Administration - EIA, 2024). From natural gas production sites, methane emission rates are estimated to be 830 Mg/h, with a high fraction from super-emitting sites (Omara et al., 2018).



Climate policy solutions generally rely on "bottom-up" inventories, which are derived from known emission rates for individual processes at a source. Bottom-up inventories are often at odds with top-down measurement techniques, which rely on measurements of atmospheric concentrations, and some have tried to reconcile these differences (Allen, 2014; Etiope and Schwietzke, 2019). Atmospheric measurements have expanded with the advances of satellite monitoring systems (de Gouw et al., 2020; Jacob et al., 2016). Both bottom-up and top-down methods have relatively large uncertainties, especially when diverse sources of methane overlap (Allen, 2016). Top-down methane emissions measurements rely on the accuracy of the instrumentation aboard the satellite, the retrieval method, as well as the methods used to calculate emissions from the column densities, such as a Bayesian inversion or a flux divergence method (Liu et al., 2021; Zhang et al., 2020). Improved methane inventories are invaluable to regulators and policymakers. Understanding the extent of methane emissions would help climate policymakers set more accurate and achievable goals and would allow regulators to effectively monitor those goals.

The accuracy of the inventory depends quite significantly on the accuracy of the measurements. The TROPOspheric MOnitoring Instrument (TROPOMI), an imaging spectrometer aboard the Sentinel 5 Precursor satellite, is known to have significant biases in the operational methane retrievals related to surface albedo (Lorente et al., 2021). There have been several recent updates to the dataset to mitigate this albedo effect using TROPOMI retrieval data over areas without emissions, and also by comparison with proxy retrievals from GOSAT, which are unaffected by surface albedo (Balasus et al., 2023; Lorente et al., 2021). The currently used albedo corrections (based on Lorente et. al.'s work) are based on a long-term correction and are best applied to annual or longer-term datasets. When applied to methane retrievals on a seasonal basis, some residual albedo effects are still apparent and may thus bias seasonal data. This study attempts to develop a seasonal albedo correction for the area of the Denver-Julesburg (DJ) basin in Colorado to account for these effects.

Colorado ranks in the top 10 states in total energy production (U.S. EIA, 2020). The state produced nearly five-times more crude oil in 2022 than in 2010 largely due to the expansion of horizontal drilling and hydraulic fracturing (U.S. EIA, 2018, n.d.), and production of natural gas has more than doubled since the year 2000 (U.S. EIA, n.d.). The majority of crude oil produced in Colorado comes from the Niobrara shale formation, located mostly within Weld county, while the whole basin stretches from southern Colorado to Wyoming and from the front range uplift into Nebraska and Kansas. (Pétron et al., 2014; U.S. EIA, 2023). Weld County is also one of the richest agricultural counties east of the Rocky Mountains producing over 27% of the entire state's agricultural sales. 80% of the land area in Weld county is used for agriculture, with 44% of that land used for cropland, and 53% used for pastureland (United States Department of Agriculture (USDA), 2017). Agriculture complicates the measurement and attribution of methane emissions data in two major ways: 1) cropland seasonal albedo shifts are not currently compensated for in albedo corrections, and 2) unreported methane emissions from animal feedlots like Confined Animal Feeding Operations (CAFO)s occur in close proximity to oil and gas production. A seasonal albedo correction would assist with both of the issues by 1) correcting the seasonal shifts in albedo, and 2) allowing for more accurate top-down seasonal methane emissions quantification, which may allow deconvolution of consistent O&G emissions from seasonal agricultural emissions. Collocation of large oil and gas production with massive agricultural operations makes the DJ basin and Weld county in particular a prime target for a machine-learning based seasonal albedo correction.



Machine learning is a subfield of artificial intelligence where a machine imitates a human brain in an attempt to solve a problem presented to it. Some machine learning algorithms have been criticized in the past as being a "black box" approach where output data is produced but the algorithm does not and sometimes cannot clarify how the output data were calculated (Rudin, 2019). With the use of more transparent types of machine learning algorithms and the help of additional python modules such as SHapely Additive exPlanations (SHAP), clearer reasonings for output can be determined (Lundberg and Lee, 2017). A neural network is a kind of machine learning algorithm that more closely resembles the human brain, containing 'neurons' that make connections and associations with various weights between themselves to determine the output values (Martín Abadi et al., 2015). Others have used machine learning previously to develop TROPOMI retrieval correction methods (Balasus et al., 2023).

Albedo corrections for the TROPOMI methane data have been described in the literature, with the most prominent being from Lorente et. al. which is now incorporated into the TROPOMI retrieval algorithm. Another well-formed albedo correction is from Balasus et. al. which also utilizes machine learning. Lorente et. al. used a B-spline interpolation of albedo dependence calculated over 2 years of data, while Balassus et al. trained a machine learning model on global data over a multi-year timescale, using the University of Leicester (UoL) Greenhouse Gases Observing Satellite (GOSAT) proxy retrievals as the target data (Balasus et al., 2023; Lorente et al., 2021). In this work we demonstrate that seasonal or monthly averaged methane retrievals continue to be biased by albedo effects after the implementation of these correction algorithms. A major reason for using a seasonal or finer time resolution average is for deconvolution of agricultural emissions. Presently, oil and gas operations are required to report on their emissions, but the accuracy is disputed (Zavala-Araiza et al., 2015). Meanwhile, agricultural operations are largely exempted from emissions reporting. The difficulty arises when agricultural and oil and gas operations are near to each other or co-located. Satellite methods for measuring emissions from oil and gas operations are going to be biased by the unaccounted-for agricultural operations. Deconvolution of oil and gas emissions, which largely remain constant through the seasons, and agricultural operations, which cycle through the seasons, could be made more accurate if the measurements could be seasonally resolved. A seasonal albedo correction, as presented here, is a step towards making a seasonal measurement more accurate for better determination of emissions.

## 2 Methods

The two satellites used in this study, TROPOMI and GOSAT, have different spatial resolutions both in a latitude-longitude grid, but also vertically with different numbers of averaging kernels. Because of this, as well as other instrument sensitivities, we don't expect TROPOMI and GOSAT to measure the same concentrations over the same places at the same time. Δ(TROPOMI – GOSAT) is an adjustment made here to place TROPOMI and GOSAT data onto common averaging kernel sensitivities and vertical profiles and determine the difference between the measurements on the same spatial scale. The calculation of this value is described in Balasus et. al. and involves interpolating GOSAT vertical pressure levels to TROPOMI's vertical pressure grid in order to calculate what GOSAT would have retrieved with TROPOMI's vertical



sensitivity ((Balasus et al., 2023). This value is used as the target of the Machine Learning (ML) models training. The predictor variables were normalized using z-score normalization to ensure the predictor values are on the same scale for training purposes.

## 2.1 Satellite Data

### 2.1.1 TROPOMI

TROPOMI is the push-broom imaging spectrometer aboard the European Copernicus Sentinel 5 Precursor (S5P) satellite, capable of measuring methane among other chemicals. It has been described in detail previously (Levelt et al., 2022; Veefkind et al., 2012). In this work, TROPOMI orbit files from April 2018 – December 2022 were downloaded from the ESA Copernicus open access hub. We used level 2 reprocessed and offline version 2.4 methane column data, XCH4, with QA values of >= 0.5, and the SWIR surface albedo as co-retrieved with XCH$_4$. The bounding box for machine learning training data used was Latitude: 34°N to 42°N, Longitude 106°W to 95°W, which encompasses the largest production regions of the Denver-Julesburg basin and extends into the surrounding states that also contain parts of the basin; Wyoming, Nebraska, and a small part of Kansas.

A well-known artifact in methane retrievals from TROPOMI is striping caused by small differences between across-track pixels, which can be mitigated by performing a stripe correction (Liu et al., 2021).

### 2.1.2 GOSAT

The University of Leicester Full-Physics dataset (UoL-FP) proxy retrieval scheme was used (Parker and Boesch, 2020). The proxy retrieval involves retrieving the CO$_2$ column to act as a proxy for aerosol scattering effects (Schepers et al., 2012). This dataset has been used extensively before as a measurement that is less affected by changing surface albedo (Balasus et al., 2023; Lorente et al., 2021). The data were downloaded from the Center for Environmental Data Analysis for the years 2018 – 2020 on a global scale and the code calculating the co-location of TROPOMI and GOSAT data to calculate TROPOMI-GOSAT pairs was based on that of Balasus et. al. (Balasus et al., 2023).

## 2.2 Machine Learning Methods

A neural network is a system of neurons, similar to a brain, with the capability to learn (Kriesel, n.d.). Here it is taught using training data with known correct answers (supervised learning), to generalize and associate data. A neural network can find reasonable solutions to similar problems of the same class that were not explicitly trained. Here, a neural network was trained on a large subset of co-located TROPOMI and GOSAT data gridded to a 0.1°x0.1° latitude/longitude square grid, totaling 17,634 points with 31 variables, described in **Table S1**, each to develop a hybrid TROPOMI/GOSAT dataset, which combines the measurement accuracy and lack of albedo effect of the GOSAT proxy retrieval with the data coverage of TROPOMI. The variables were selected based on previous ML work on this topic with some minor changes (Balasus et al.,



2023). We chose to incorporate the retrieved and corrected XCH₄, and we chose to remove the surface classification variable because our relatively smaller area of study has relatively few bodies of water. Furthermore, we chose to remove wind speed

variables so that we would not introduce a bias or double counting if this model were to be used with the flux divergence method of quantifying methane emissions, which requires wind speed and direction (Beirle et al., 2021). A neural network can be described as "deep" if it has 3 or more "hidden layers" or levels in the network. The term "Transfer learning" is used to describe a model that has been trained previously and is subsequently trained again starting from the previous training endpoint. A Deep Transfer Learning (DTL) method was used where an annual base model was trained and tested on 80% of the total

points. These same points were then separated by the month of their collection, and used to train 12 separate monthly models, starting from the annual base model; a schematic representing this training process is presented in **Fig. 1**. DTL is especially suited to this type of learning because 1) the initial learning phase trained on all training data helps the lower levels of the model learn to generalize the task, and 2) the subsequent training occurs on much smaller monthly training datasets that help train higher, more specific levels of the model. Various hyperparameters were tuned using Optuna, a hyperparameter tuning

package for Python (Takuya et al., 2019).

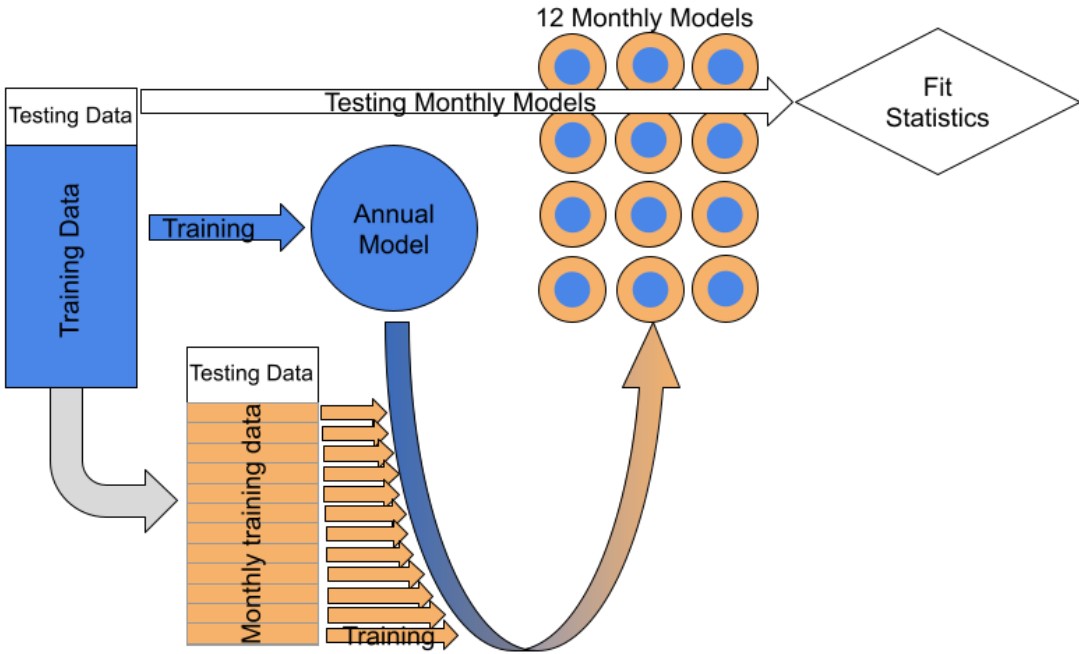

**Figure 1. Schematic representation of the data training process. Blue represents the annualized, long-term model, while orange represents the short term monthly-seasonal model and data. Transfer learning, the process by which a pre-trained model is trained again, usually on more specific data, was utilized here to generate 12 monthly models with the deeper understanding that comes from larger data quantities in the annualized model combined with the better specialization of the monthly-seasonal training data,**

**represented by the orange circles with the blue center.**



## 2.3 Python

All of the computations were completed using Python, a general-use, interpreted, object-oriented, programming language ideal for building and implementing machine learning models and algorithms (Van Rossum and Drake, 2009). A number of third-party packages were useful in the computations completed in this work. Matplotlib for figure generation (Hunter, 2007); the machine learning packages: TensorFlow (Martín Abadi et al., 2015), Keras (Chollet and others, 2015), XGBoost (Chen and Guestrin, 2016), LightGBM (Ke et al., 2017), and Catboost (Dorogush et al., 2018); Pandas and Geopandas for tabular and geospatial data organization (Jordahl, 2014; McKinney, W. and others, 2010); Optuna and Fast and Lightweight AutoML Library (FLAML) for tuning ML models hyperparameters (Takuya et al., 2019; Wang et al., 2021); Scipy for scientific and statistical functions (Virtanen et al., 2020); Shapely for manipulation of geometric objects (Gillies and others, 2007); Numpy for array manipulation (Harris et al., 2020); netcdf4 for opening and reading satellite data (Whitaker, 2008); Rasterio for raster manipulation (Gillies et al., 2013). and tqdm to visualize data processing progress (da Costa-Luis and Yorav-Raphael, 2013). Figure 1 was created using the Google drawing suite, Fig. 2 was created using Python matplotlib, Figs. 3-7 were created in Igor Pro 8.04.

## 2.4 Other Geospatial Data

River paths and extent data for the South and North Platte Rivers were downloaded from NOAA (Rivers of the U.S., 2024). Crop data were downloaded from CropScape, a geospatial thematic agricultural mapping software (Han et al., 2014). Cartographic shapefiles containing state, county, and urbanized area boundary lines were downloaded from the US Census Bureau (Cartographic Boundary Files - Shapefile, 2024). Finally, data visualizations were made to be color-accessible by Fabio Crameri's scientific color maps (Crameri et al., 2020).

## 3 Results and Discussion

The seasonal biases of the current TROPOMI operational product, which includes the albedo correction from Lorente et al. (2021), are studied in **Fig. 2** for the area of interest. **Figure 2** shows the ratio between co-located GOSAT and TROPOMI methane retrievals as a function of surface albedo in the short-wave infrared (SWIR). In the ideal case, these ratios are equal to 1 and there is no correlation between this ratio and surface albedo (R=0). When all data are used (**Fig 2a**) the Pearson correlation is indeed calculated to be low, i.e. below a threshold of 0.1, which we chose here as a target value for minimal correlation between SWIR surface albedo and the albedo corrected methane retrieval. When the data are shown by season, this is no longer true - Pearson correlations with an absolute value greater than 0.1 indicate that there exists some correlation between the SWIR surface albedo and the albedo-corrected methane retrieval. This is not entirely surprising, as the built-in albedo correction is not designed to be seasonal and was calculated for use across the entire dataset, and not our small subset in Colorado. **Figure 2b&c** demonstrate the change in surface albedo as a function of season, with the density of counts shifting from the left side of the plot, indicating smaller albedos, to the center of the plot, indicating higher albedos on average from





summer to winter. The different seasons also have different directions of change, with summers being an inverse correlation
and winters being a positive correlation. The QA value used in processing the TROPOMI retrieval data retained high-quality
snow-covered scenes, so some of this shift could be attributed to the SWIR reflectance of snow over bare soil. Regardless of
180 the reason, the shifting albedo and seasonally variable albedo effect biases methane retrieval data from TROPOMI at finer
time scales. In order to correct for this bias we employed a DTL neural network machine learning algorithm.

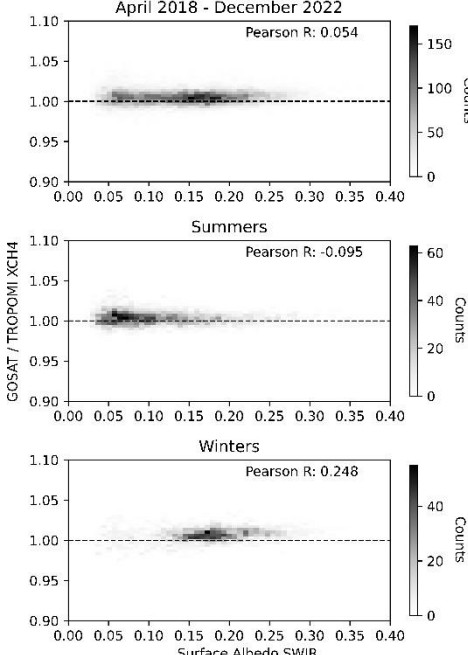

**Figure 2. Albedo effect on methane retrievals on seasonally-averaged TROPOMI data. TROPOMI bias corrected methane level 2
retrieval data averaged from April 2018 to December 2022 (a) All months, (b) summer months (July-Sept.), and (c) winter months
(Jan.-Mar.). The TROPOMI data are co-located in space and time with UoL GOSAT proxy retrievals treated as ground-truth. The
185 dashed line represents perfect overlap and no correlation. Pearson R values represent the correlation between surface albedo and
XCH4 retrievals. Our target Pearson R values are -0.1 < R value < 0.1.**

## 3.1 Model Evaluation

The DTL neural network models were trained and evaluated as described in **Sect. 2.2** and compared against the
uncorrected methane retrieval, the Lorente et. al. methane corrected retrieval, and the blended TROPOMI/GOSAT product
produced by Balasus et. al., commonly referred to as "Harvard dataset", for their effectiveness in methane correction. To
evaluate against the other models, Pearson correlations were calculated and presented in **Fig 3a** with uncorrected data as x =
surface albedo SWIR as retrieved by TROPOMI and y = raw co-located proxy retrieval GOSAT data / TROPOMI $XCH_4$, for
the Lorente et al. correction, y = raw co-located proxy retrieval GOSAT data / TROPOMI bias corrected $XCH_4$. Because the
Harvard dataset and the data in this work both used sensitivity adjusted GOSAT proxy retrieval data in the calculation of the
195 corrected TROPOMI data, the same sensitivity adjusted GOSAT data are used to determine the Pearson value, with x = surface
albedo SWIR as retrieved by TROPOMI, and y = sensitivity adjusted co-located proxy retrieval GOSAT data / TROPOMI





corrected $XCH_4$. **Figure 3b** depicts the 95% confidence intervals about the mean of the 12 months of the Pearson values and is helpful in determining the most effective model. Dashed lines in both figures represent the ideal values indicating no correlation (Kuckartz et al., 2013), with values in **Fig. 3a** being between the dashed lines at -0.1 and 0.1 Pearson correlation value, and **Fig. 3b** being the average and center of the 95% confidence margin of error on the line at 0 Pearson correlation value.

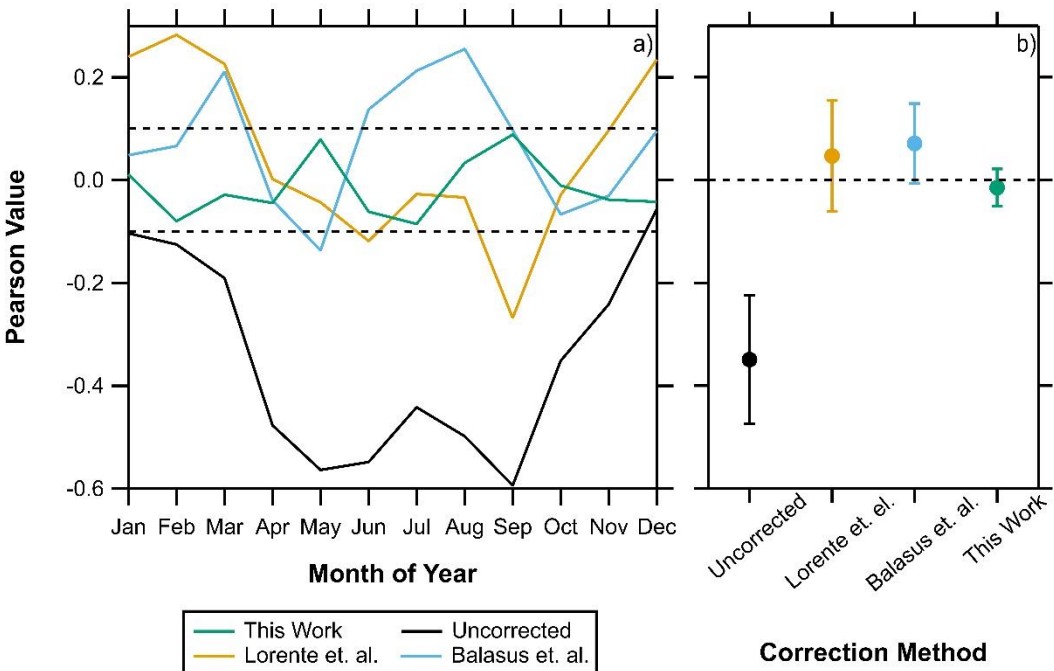

**Figure 3. Comparison of the model developed in this work with Lorente et. al., and Balasus et. al. corrections, and uncorrected TROPOMI retrieval data. Pearson value describes the Pearson correlation value of (sensitivity corrected GOSAT / measured or calculated TROPOMI value) and surface albedo SWIR for the ML model predicted data, the Pearson correlation value of (raw GOSAT / calculated value) and surface albedo SWIR for scalar corrections, and the Pearson correlation value of (raw GOSAT / raw TROPOMI value) and surface albedo SWIR for the uncorrected data. (b) points represent the average and error bars describe 95% confidence intervals of the 12 months.**

Only the models devised by this work entirely remove the seasonality described by the uncorrected data. Additionally, the Pearson values remain within our goal 0.1 Pearson correlation value for each month. As expected, the uncorrected data reach the farthest outside of this range and remain outside for the greatest number of points. The Lorente et. al. correction, which was not designed to handle seasonality, significantly improves upon the uncorrected data, but preserves the seasonal trend in the data, demonstrating larger, positive correlations in the winter months and cycling through the seasons. The Balasus et. al. blended dataset improves this further by reducing the seasonality of the correlation, but this dataset still displays correlations outside the 0.1 correlation threshold desired. Finally, this work's devised monthly models always fall between the ideal -0.1 and 0.1 Pearson values. In comparing the mean and 95% confidence intervals, we observe the steady improvement in the



progression of models, with this work's monthly models providing for the Pearson value closest to 0 and with the smallest 95% confidence interval. All this demonstrates that the use of the monthly models provides a small, but measurable improvement over previously designed models for albedo correction in this specific region around the Denver-Julesburg basin.

## 3.2 Model Results

The python library SHapely Additive exPlanations (SHAP) was used to determine the relative importances of the different variables incorporated into the model (Lundberg and Lee, 2017). The importance of a variable indicates how much each variable contributes to the difference between the actual model output and the average model output. The importances of the variables were calculated on a monthly basis to show how the importances change over time, and two representative months, one for winter, one for summer, are shown in **Fig. 4**. **Figure 4a** depicts the model outputs for the month of January in

a decision plot. Decision plots are generally used to show how models make their determinations and what variables are affecting their decisions the most. Here the decision plot is showing that the range of correction values stretches from approximately -40 to 40 ppb, indicating small changes in the total methane concentrations (~2-4%), this change is larger than the mission specifications of bias less than 1.5%, and much larger than the measured mean bias of the corrected TROPOMI XCH$_4$ data of 0.2% (Apituley et al., 2022; Landgraf et al., 2023). That the corrections are larger than the biases suggests that

the corrections are significant and important. Contrasting the general shapes of the decision plots, **Fig. 4a** appears to be more cone shaped, having a much starker taper in the less important variables, while **Fig. 4b** appears more cylindrical, sporting a milder taper. This indicates that the relative importance of predictor variables changes between seasons. A single model would miss this detail entirely, but the set of 12 monthly models allows for this change to occur. Additionally, the final model output value for **Fig. 4b** remains in the same range of approximately -40 to 40 ppb. Together, this indicates that while outputs remain

in the same range, the difference in the importance of the variables changes the method that the models use to predict the outcome.

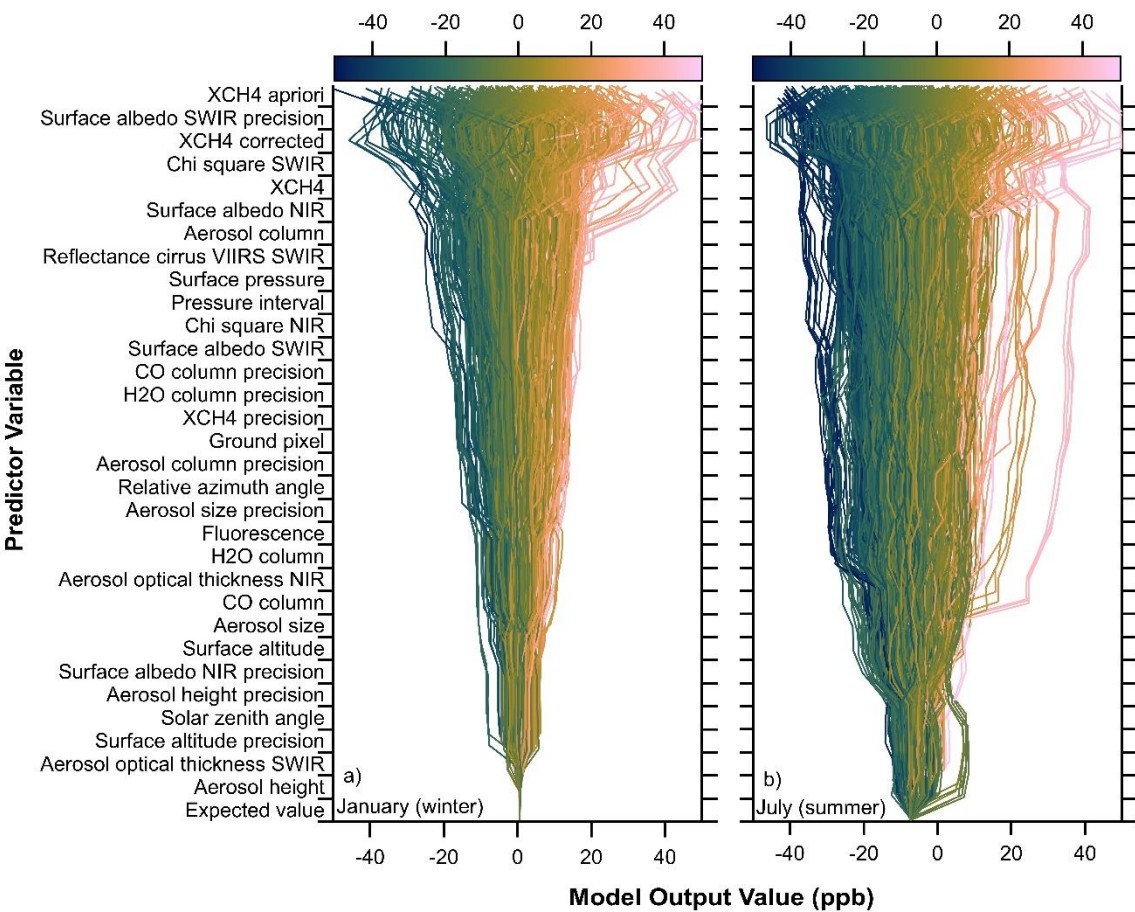

**Figure 4. Decision plots depicting relative importances of predictor variables on a seasonal basis. SHAP importances were calculated for a) January and b) July and the contributions from each predictor variable are shown. Variables are ordered from top to bottom by importance in January. Color scale indicates the final model output value, which is the Δ(TROPOMI – GOSAT) value. Expected value is the average prediction made by the model across all possible combinations of features, and is thus the same value for all trials using the same model.**

While the training process attempted to minimize the differences between TROPOMI and GOSAT data, thus effectively reducing the dependence on SWIR surface albedo, not all training iterations were successful in this due to the multitude of features to incorporate. As part of our model validation, we only considered those that reduced the correlation between XCH4 and surface albedo SWIR as viable models. Due to this validation method, we call our machine learning product an "albedo correction". **Figure 4** shows that other features may be more important than the surface albedo SWIR in the actual model calculation. The variables that appear higher on the y-axis than "surface albedo SWIR" tended to be more important and should be analyzed as well. Some of these variables have clear reasonings as to why they are more important: XCH4 apriori, XCH4 corrected, and XCH4 are all the measurements of methane mixing ratio that were either priors for the TROPOMI measurement (XCH4 apriori) or direct measurements of the methane mixing ratio by TROPOMI (XCH4 and





XCH4 corrected). XCH4 and "XCH4 corrected" directly measure methane mixing ratios via TROPOMI, serving as primary data sources for our predictive models. The reasoning for other important variables is not so clear: "surface albedo SWIR precision" and "chi square SWIR". The precision of the surface albedo SWIR measurement being important was not expected, but may be the result of a well-trained model successfully making the association between the SWIR albedo measurement and its precision. A less precise measurement would be less heavily relied upon for the model's predictions, so the importance may come from the association between the precision measurement and how much a particular measurement affected the model during training. Similarly, the "chi square SWIR" is a goodness of fit check that ensures that the SWIR measurements by the instrument fall within an appropriate distribution. Poor goodness of fit could allow the model to rely less heavily on that particular training data point in making future predictions. Additionally, there were some factors that appear lower on the y-axis that are somewhat unexpected, such as aerosol optical thickness SWIR and solar zenith angle. Aerosol optical thickness SWIR describes the atmospheric density of aerosols that reflect in the SWIR band, which could be expected to be important for this prediction due to the importance of the other factors affecting the SWIR band that appear towards the top of the axis. Solar zenith angle is a fundamental factor in the calculation of the methane mixing ratio because it describes the angle of incident light, which is integral to remote sensing by satellites. That this factor is relatively unimportant suggests that this information is well incorporated in the retrieval. This study utilized an implicit stripe correction instead of an explicit one. The UoL target data are not subjected to a striping effect, so the use of the target data and using the ground pixel index as a variable in the model allowed for a stripe corrected dataset to be output from the input of non-stripe corrected data. This process relies heavily on the 'ground pixel' variable which finds middling importance in **Fig 4.** indicating that while the stripe correction is important, other factors affect the overall output more. Other information describing the training and validation process is available in the supplemental information.

### 3.3 Model corrections in practice

The trained models were then used to predict corrected XCH4 values on a monthly basis on data from April 2018 to December 2022, the correction values for which are depicted in **Fig. S1**. The months of January and July, representing winter and summer data respectively, are presented in **Fig. 5**. The model predicted positive and negative correction values for this data appears to be seasonally dependent, with more positive corrections being made in colder months and negative corrections being made in warmer months, appearing as blue colors in the summer (**Fig. 5a**) and brown colors in the winter (**Fig. 5b**). The correction values also show a specific geographic distribution; two curved lines, one curving upwards from Denver, the other curving down through Nebraska appear to follow the South and North Platte Rivers respectively (white dashed lines in **Fig. 5a**). As Colorado has been described in the past as part of the "Great American Desert," water sources like these two eventual tributaries to the Missouri River dictate where larger water-intensive agricultural operations exist. As such, larger densities of water-intensive crop farms are co-located with these rivers, bringing their albedo-influencing crops and plant-life, and thus requiring an albedo correction which is not necessarily reflected in magnitude by the surrounding scrubland. It has been shown that water intensive crops, like corn, sugarbeets, and alfalfa; and drought resistant crops, like winter wheat, millet, and dry



beans; reflect SWIR light differently, allowing for identification of crops from space with the SWIR reflectance variable along with other variables (Chen et al., 2005). This effect is possibly due to water content or leaf size of the vegetable matter. The spatial extent of the water intensive crops is much wider than the riverbed; the North and South Platte rivers are extremely small (average discharges 1,355 and 175 cu ft/s respectively, Mississippi river is 593,000 cu ft/s) and are far less in extent than one satellite pixel. In his book *Roughing It*, Mark Twain describes the South Platte in 1870 as "shallow, yellow, muddy… and

only saved from being impossible to find with the naked eye by its sentinel rank of scattering trees standing on either bank." (Twain, 1891) **Figure 5** shows how the XCH₄ correction factor changes in different ways across the seasons for water intensive and drought resistant crops. Comparing these changes with the unfarmed grassland in the same region, the effects are small but substantial. In the coldest of winter months the river structures appear less prominently, if at all.

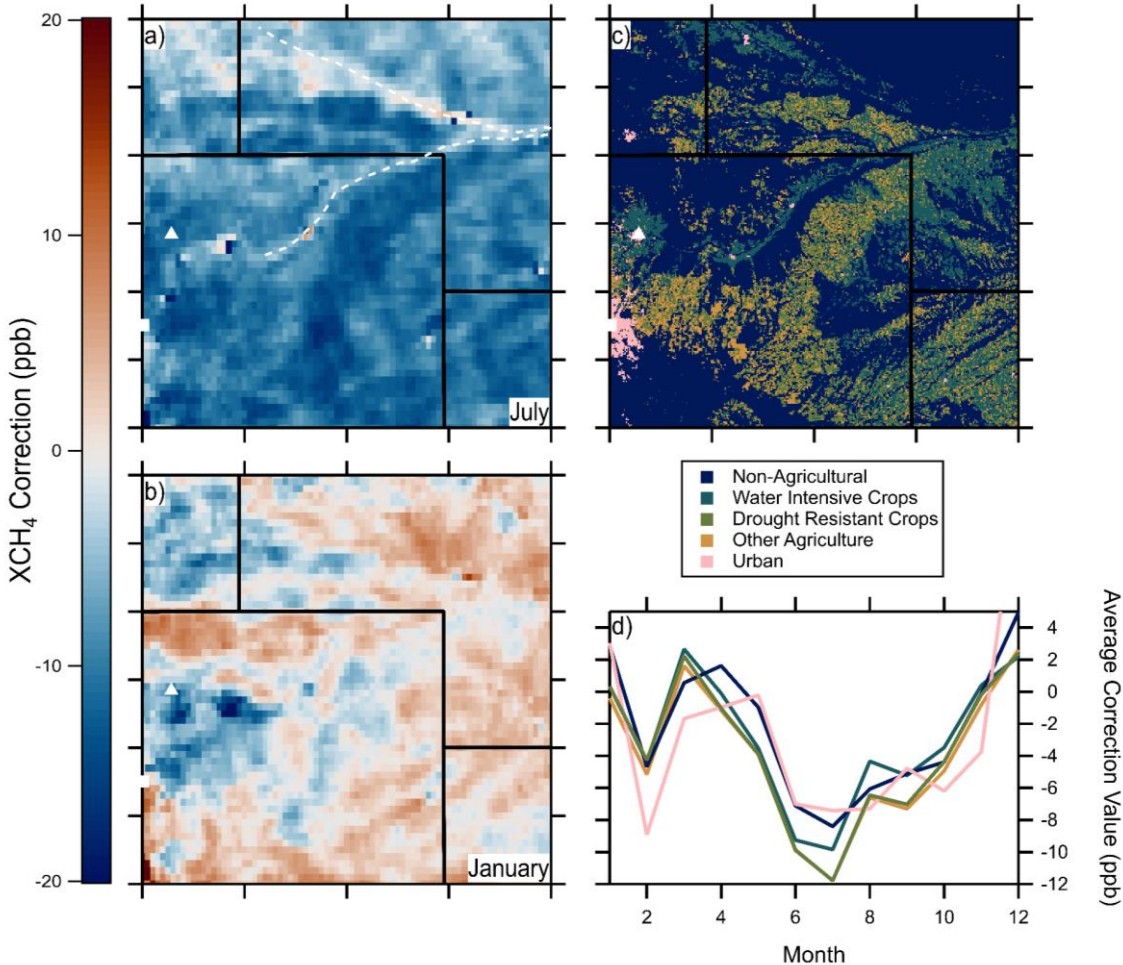

**Figure 5. Average XCH4 correction values for water intensive vs. drought resistant crops. XCH₄ correction value maps with white**
**dashed lines representing the North and South Platte Rivers for July (a), and January (b) representing data for the summer and winter respectively. Locations of crop types (c) around the DJ basin. Water intensive crops include corn, alfalfa, and sugarbeets, drought-resistant crops include winter wheat, millet, and dry beans. Crops with both traits, fallow land, and other agricultural types**




are described as Other Agriculture. Grassland and other non-agricultural types, except urban areas, are described as Non-Agricultural. Developed Land includes parts of the cities of Denver, Greeley (marked with the white square and triangle respectively), Cheyenne, and other smaller communities. Average CH₄ Correction values for the crop types (d), and Drought resistant crops require larger corrections throughout the summer months while water intensive crops are more similar, though not the same, as the surrounding grasslands. No error bars are shown due to the large amounts of points making both standard error and 95% confidence interval values too small to see. Crop data is from 2021 only and calculated using the April 2018 – December 2022 correction data.

Particularly prominent in **Fig. 5a** is a darker swath south of the upward bend in the South Platte river. This area also has many farms, but these farms are more likely to grow drought-resistant crops. Additionally, many more of these fields lie fallow in a given year than the ones irrigated by river water. Another area of agricultural significance is around Greeley, Colorado (white triangle), named for, and partially funded by, Horace Greeley, a 19th century newspaper editor who urged the nation to seize the opportunity to convert the desert to cropland (Reisner, 1993). Greeley is also visible in the colder months maps, giving further indication that cropland is associated with albedo effects, but with magnitude or direction differing based on crop types and growing seasons. Greeley and the surrounding farms make up a large portion of the crop farming capacity within Weld county.

**Figure 5c** depicts the agricultural land use in the area of interest where visual comparison of the water intensive crops and the bright-line regions of the summer seasonal albedo correction plots can be made. Numerical comparison agrees with visual inspections, as **Fig. 5d** depicts average albedo correction values over each kind of land cover. Overarching seasonal trends appear, with corrections over all land covers appearing closer to 0 in the winter and fall and increasingly negative through the spring and summer. Additionally, seasonal effects over individual types of land cover are measurable. During the winter and fall, many of the land cover types appear very similar, while diverging from each other in the spring and summer, when vegetation in Colorado becomes increasingly stressed for water.

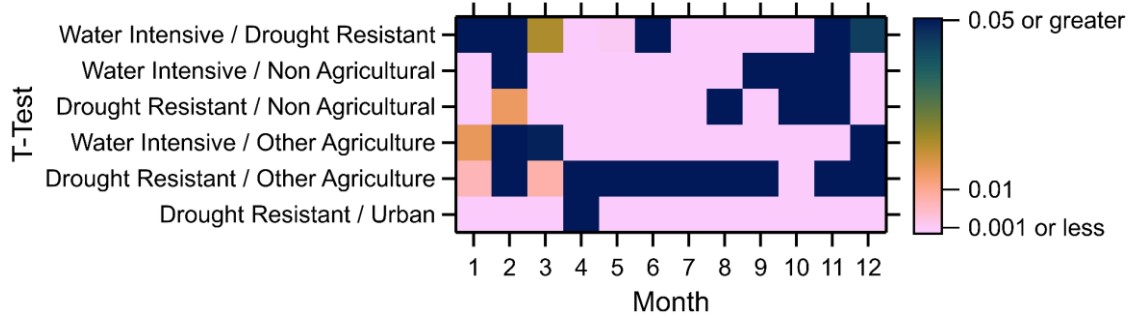

**Figure 6. Significance tests demonstrating the statistical significance between paired datasets. All values that are not the darkest blue = 0.05 or greater are significantly different in a p critical = 0.05 environment. All values that are pink = 0.001 or less indicating very significant differences. More blue near in the early year and later in the year indicate that albedo corrections are more similar between different land cover types, and more pink in the summer months indicate that albedo corrections are more different between different land cover types in this time period.**

T-tests were performed between categories to determine the significance of the differences between the different land uses and presented in **Fig. 6**. T-tests for each month of data on a small subset of 500 points for each land use demonstrate, for example,





that drought resistant crops and other agriculture types are not statistically significant. P values for the T-tests between other land uses tend to increase and indicate no statistical significance in the winter and late fall, while indicating statistically high significance throughout the spring and summer for most land use pairs for most months. This indicates that in general the

different land uses require different correction values and this is related to the kinds of agriculture utilized. Water intensive agriculture is likely irrigated and soil moisture and vegetable water content can play a significant role in surface albedo SWIR, such that measurements of the like have been used to measure extents of irrigated agricultural land uses (Chen et al., 2005). This demonstrates that a seasonal albedo correction is important and may be different in different parts of the world over different land cover types. Similarities between water intensive and non-agricultural and drought resistant and non-agricultural

in the winter and fall indicate that non-agricultural land may not be as affected by the seasonal bias, thus confirming our suspicion that seasonally changing farming practices may be the root cause for the albedo effect issues.

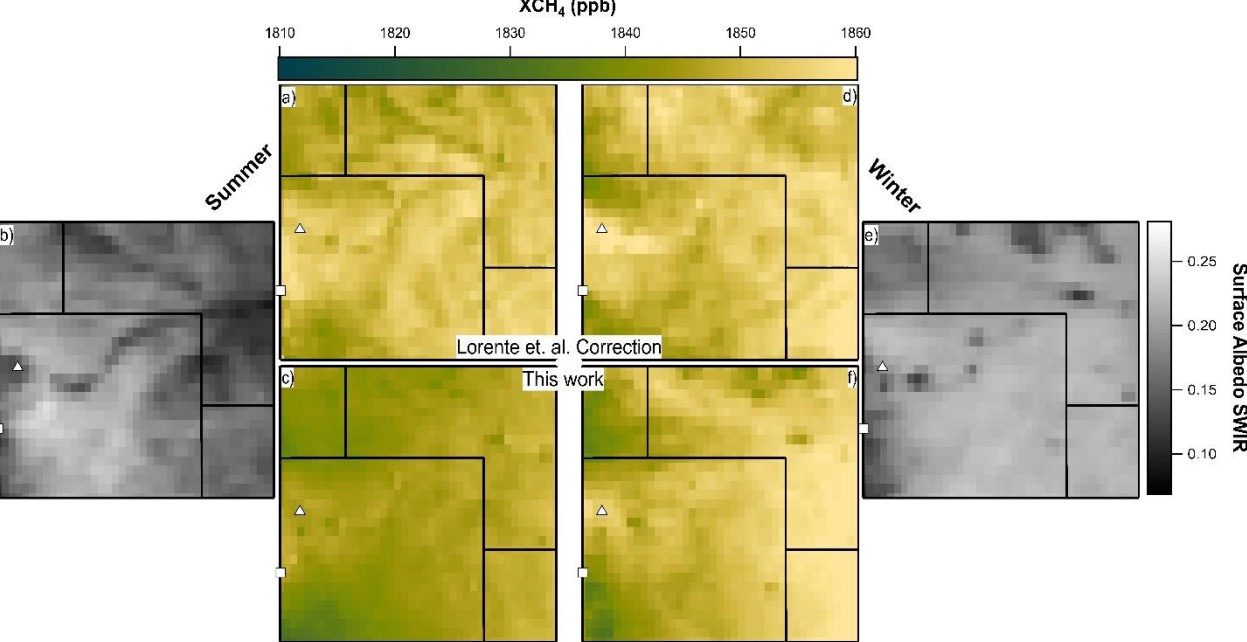

**Figure 7. Result of the methane albedo correction devised in this work. The Lorente et. el. correction in the summer (a) appears to have similar features as the surface albedo SWIR retrieval over this same area (b), while the features have been removed in the**
**correction devised by this work (c). In the winter, the surface albedo SWIR product (e) appears quite different than in the summer, indicating the need for this seasonal albedo correction. The Lorente et al. correction (d) and the correction devised by this work (f) appear more similar in the winter.**

The corrected XCH$_4$ data were calculated and averaged across the summer and winter months to demonstrate the difference the models developed here make in their corrections. Visually apparent in the Lorente et. al. corrections in **Fig 7**

**(a&d)** are structural features that are similar to features shown in the surface albedo SWIR retrieval (**Fig. 7 (b&e)**). The corrected dataset devised here appears to have slightly a reduced mixing ratio, appearing slightly darker in color than the





Lorente corrected data. This is likely due to the correction's dependence on the UoL GOSAT proxy retrieval target data, which on a global average measures 9.2 ppb less XCH$_4$ than TROPOMI (Balasus et al., 2023). Notably, the Denver metropolitan area appears to have lower methane concentrations in our model output data than in the original TROPOMI Lorente et al. corrected data. The corrected data (**Fig. 7 (c&f)**) appear visually smoother, which is to be expected for a long-lived analyte like methane. **Figure 7** cannot be evaluated as before with a Pearson correlation because the correlation requires GOSAT / TROPOMI data to be used to account for natural correlation between surface albedo SWIR and XCH$_4$. There is not sufficient GOSAT data over this extent and time period to calculate such a Pearson correlation. Instead we assume that the tested model output correlations hold for this data, making the correlations between GOSAT/TROPOMI and the surface albedo SWIR: -0.03±0.04 and 0.01±0.08 for winter and summer respectively for the models developed in this work; and 0.25±0.03 and -0.1±0.1 for the Lorente et al. correction values; error values are 1σ. Overall it appears that the correction is effective in removing the albedo effect over seasonal time resolutions.

## 4 Conclusions

A small but significant seasonal dependence on surface albedo biases was found in TROPOMI XCH4 retrievals even after the application of the current state-of-the-art albedo corrections when focusing on a single region. A series of deep learning ensemble models specifically designed to reduce differences between TROPOMI and GOSAT while also reducing dependency on surface albedo in the SWIR have been developed to improve upon previous corrections. The output of the trained models removes the seasonal dependence on surface albedo and demonstrates the fewest exceedances of a -0.1<R<0.1 Pearson correlation with surface albedo in the TROPOMI dataset. Application of the albedo correction to the Denver-Julesburg basin reveals albedo correction dependencies on land-cover, requiring larger in magnitude corrections in the summer months over drier, drought-resistant crops than irrigated water intensive crops, with differences that also fluctuate seasonally. The 12 monthly models seasonal albedo correction appears to resolve previously understudied issues surrounding long-term albedo corrections over seasonally changing areas, like cropland, making this a valuable tool for developing more accurate methane emissions inventories, models, and potentially deconvoluting relatively constant oil and gas emissions from seasonally dependent agricultural emissions. Methane measurements corrected utilizing this albedo correction method will be quantified in a forthcoming publication.

## Code availability

The code used for all portions of this project is available at https://github.com/bralex63/tropomi_seasonal_albedo_correction/tree/v1.0 and archived on Zenodo at https://doi.org/10.5281/zenodo.12809441.



**Data Availability**

The TROPOMI data used here are available at https://browser.dataspace.copernicus.eu for April 2018–present. The GOSAT data used here are available at https://doi.org/10.5285/18ef8247f52a4cb6a14013f8235cc1eb for 2009–2021. The agricultural data used here are available at https://nassgeodata.gmu.edu/CropScape/.

**Author Contributions**

AB and JdG designed the study. AB performed the analysis with contributions from JdG and BD, and led the writing of the paper with contributions from all co-authors.

**Competing Interests**

The authors declare that they have no conflicts of interest.

**Financial Support**

This work was supported by the Colorado Department for Public Health and Environment (CDPHE) grant no. 2024*2228 and the Cooperative Institute for Research in the Environmental Sciences (CIRES) Graduate Fellowship.

**Acknowledgements**

The authors would like to thank Ilse Aben, Ben Hmiel, and John Evans for useful discussions, and Bud Pope for the cooperation
with Blue Sky Resources. They would also like to thank Raquel Serrano-Calvo for her unpublished results that inspired us to perform this analysis. This work contains modified EU Copernicus Sentinel-5P TROPOMI data (2018-2024).

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
