# Peer review of "Deep Transfer Learning Method for Seasonal TROPOMI XCH4 Albedo Correction"

_EGUsphere, 2024_

## Referee Comment (RC2)

**Review of "Deep Transfer Learning Method for Seasonal TROPOMI XCH4 Albedo Correction" by Bradley et al.**

This manuscript describes the use of a deep transfer learning (DTL) method for applying bias correction to TROPOMI XCH4 retrievals over a region of the central U.S. The study develops on previous bias-correction schemes for TROPOMI $CH_4$ through use of DTL in order to produce monthly-varying models for the bias correction. The selected target region covers parts of Colorado and some surrounding states, areas with wide-ranging agricultural activity, urban areas, and undeveloped rural areas, providing good testing data for their models.

Overall, the authors have done a good job of developing and applying their model to the selected region and show quite well the impact of the monthly-varying models on the imposed bias corrections, linking them to different crop types and other landscapes. This has the potential to be of some interest to the wider community. However, in my opinion the manuscript requires some significant work before it is ready for publication.

If the authors can adapt the manuscript based on the following comments, then the work can be published in AMT.

**Major comments**

1) Throughout the paper, the authors are muddled in their explanations and terminology about the seasonality of the albedo correction in their work and the previous works by Lorente et al. and Balasus et al.. As I understand it, in both of these previous works the corrections applied to the retrieved XCH4 do vary over time (as albedo and other variables change), it's just that the models used for calculating these corrections use all available data and do NOT vary with time. Bradley et al. are quite careless in their terminology throughout in describing this concept and it is misleading and confusing for the reader.

2) The above point raises important (linked) questions that Bradley et al. do not satisfactorily discuss in the manuscript:
- What is the physical justification for applying a monthly-varying bias correction model as opposed to a fixed model? I assume that it's ostensibly that the relationship between the satellite bias and the predictor variables changes with time, but why is this exactly?
- The monthly DTL models appear to assign different relative import to the input variables in the different seasons, but why is this the case, and what is the implication of this?
- Why exactly is the temporally-fixed model of Balasus et al. for example, not sufficient in this case (beyond the fact that it is trained on global rather than regional data)?

   These questions must be raised and discussed in my opinion. As a non-expert in ML methods, I can't comment on the technical detail, but in other data-fitting

schemes there is the concept of 'overfitting' one's model to the 'true' or observational data. Is there a similar possibility here, where applying the DTL models separately to each month's data might destroy the conceptual generality of the model?

3) The authors here build on and compare to Balasus et al.'s global correction model and they have changed 3 major factors in comparison to that model – the regional bounds covered by the training data; the use of the DTL method for monthly models; and finally, the chosen subset of predictor variables. Therefore, it's impossible to say how much of the improvement is due to the specific regional training and DTL method, and how much is simply due to the fact that (e.g.) XCH4 and XCH4_corrected are now input variables (and other variables have been removed). Is it possible to include results from a version of your regional DTL models that keeps the same predictor variables as Belasus et al.?

4) The inclusion of XCH4 and XCH4_corrected is not justified in this work and should be. The authors state that they stopped using wind speed variables to avoid circular use of inputs in potential top-down application of this data, but shouldn't the use of XCH4 itself should be avoided for a similar reason? Especially as XCH4_apriori, XCH4_corrected and XCH4 turn out to be 3 of the 5 most important predictor variables.

**Specific comments**

Line 28: This Karakurt et al. reference is quite dated now – can you cite more up-to-date values?

Line 36: Quantify 'relatively large'.

Line 38: Note the assumptions and limitations of bottom-up methods also.

Lines 48-49: These statements about the limitations of current albedo corrections need a reference.

Line 59: Clarification needed here – "cropland seasonal albedo shifts are not currently compensated for in albedo corrections". However, as I understand it, the other correction methods should indeed account for the changing albedo (which is retrieved for each sounding from the satellite and included in their correction models) – it's the model for the magnitude of the correction which is consistent over time.

Line 76: If you want to stop including the year for Lorente et al.'s and Balasus et al.'s work from this point, you should now clearly state that the shorthand that you're using thereafter refers to a particular paper (with year cited).

Line 76: What do you mean by 'well-formed'?

Line 92: 'different numbers of averaging kernels' – Is this correct as stated? Is it clearer to say that the (number of/locations of) vertical retrieval pressure levels are different for the two satellites, and therefore the averaging kernels are different?

Line 106: Define QA.

Line 107: SWIR is defined later in the text but should be defined here (or possibly earlier?).

Line 112: Make clear here that this is not something that you do in this study, however, due to your implicit striping correction?

Line 119: Add more detail here about the co-location criteria so the reader does not have to refer to the Balasus paper.

Line 121: This Kriesel reference needs more information for the reader to find it. Indeed the author himself explicitly asks that any citation includes the URL, and states a year. http://www.dkriesel.com/en/science/neural_networks. There are other locations where the authors should be more careful and explicit with their citations.

Line 127: Are these changes really 'minor'? See earlier major comment.

Line 128: To be clear here – what exactly is the 'corrected XCH4'? The Lorente et al. bias correction?

Line 132: As a non-expert, I need more clarification of what the hidden layers/levels are.

 Line 134: More detail needed here. Trained and tested on 80% of the data? How much of this was training data and what was the other 20% used for? How was the 80% selected? Was it evenly distributed across months – or do months potentially get different amounts of training data? Would that be an issue if so?

Lines 192 – 197: I found this information a little hard to follow as structured here. Can you make clearer?

Line 211: 'not designed to handle seasonality' – rephrase.

Around line 265 – what does it mean that the relative importance of predictor variables in your work appears quite different to those of Belasus et al.? (e.g. Aerosols/Surface Albedo SWIR are very important for them).

Figure 5: Discuss – why are the urban corrections so variable? Should they be relatively constant?

Line 325: To be clear – is this testing the significance of the differences between derived corrections over the different land-types?

Line 336: I still find this reasoning confusing – I apologise if I'm misunderstanding. Other corrections do take account of the changing albedo of the crops, which is retrieved by TROPOMI, right? So the reason that they're not capturing the seasonal variability is their model? But why is the relationship between bias and albedo changing with time?

Figure 7: I think I'm right in saying that figure captions in this journal should include a description of the figure only and not 'discussion'-type text such as this one comparing results in the different panels.

Line 346: 'appears to have' – you should be able to quantify this.

Line 347: 'due to the correction's dependence on the UoL GOSAT … data' – you should confirm that the Lorente data is based on GOSAT? I was under the impression that Lorente et al.'s correction was not based on GOSAT.

Line 350: 'appear visually smoother' this is vague and unscientific.

Lines 351 – 357: I'm slightly unclear as to what you are saying with this statement regarding the lack of necessary GOSAT data. Can you clarify further? Is it that you are not able to produce a map of correlations?

**Technical corrections**

Line 35: The (number/density/range?) of atmospheric measurements has expanded...

Line 43 and elsewhere : Include hyphen:  "Sentinel-5 Precursor"

Line 47: "currently-used"

Line 51: top 10 U.S. states

Line 61: change to: (CAFOs)

Line 64 and elsewhere: be consistent in spelling of collocation/co-location throughout.

Line 275: "Model-predicted"

Line 277: Surely red, not brown? Additionally, it would be good if Figure S1 used the same red-blue colourbar if possible. The green colours (ranging from around +5 to -15 ppb) are hard to distinguish from each other.

---

## Author Comment (AC1)

*Reviewer comments are italicized*

      Author responses are indented and unitalicized

      Changes to the manuscript text are in blue

      The authors would like to thank the anonymous reviewers for their time and attention to this manuscript. Your comments and suggestions have been greatly appreciated and have significantly improved the paper.

**Response to anonymous referee 1**

*I found the results to be of some though limited interest because they show how a method previously developed for correcting TROPOMI data on a global scale can be customized for local application to improve the correction.*

      The authors would like to clarify that this is a novel method that uses similar elements to previous corrections, and it does not primarily address global vs. local differences, but rather monthly vs. annual correction values. As monthly corrections are more spatially dependent, we executed this correction on a more local scale. Theoretically this method could be scaled up, but would likely have to be executed separately on local domains and later pieced together for the most valid results.

*But I'm puzzled by the motivating premise of this paper that the TROPOMI retrieval does not account for seasonally variable surface albedo. I believe that it does because surface albedo is co-retrieved with methane for every spectrum.*

      The reviewer is correct that some seasonality is incorporated into the existing albedo correction algorithms. The albedo retrieval has high time resolution, but the basis of generating a correction factor in the operation product does not. We have gone back through the manuscript to clarify this and to explain more clearly that the seasonality that we describe correcting comes not from the incorporation of daily surface albedo measurements, but instead a changing relationship between the surface albedo SWIR (as well as other variables) and the correction value. For example: the relationship between sunlight angle, aerosol optical thickness, and the correction value is likely quite different in the summer than in the winter in a way that the static correction value curve constructed by Lorente et al. does not take into account.

      In many regions, especially those with agriculture, surface reflectance depends on season, but corrections for this dependence are calculated from long-term averages.

      When applied to methane retrievals on a seasonal basis, we show that some residual albedo effects are still apparent and may thus bias seasonal data.

      In this work we demonstrate that seasonal or monthly averaged methane retrievals continue to be biased by albedo effects after the implementation of these correction algorithms.

      When the data are shown by season, this is no longer true - Pearson correlations with an absolute value greater than 0.1 indicate that there exists some correlation between the SWIR surface albedo and the albedo-corrected methane retrieval. This correction algorithm does account for some seasonality because the TROPOMI retrieved variables include the surface albedo SWIR which is used in the Lorente et al. algorithm to calculate a correction value. The seasonal correlation reappears after the Lorente et al. correction because this correction assumes that the relationship between surface albedo SWIR and correction value is static over time.

Regardless of the reason, the shifting albedo and seasonally variable albedo effect biases methane retrieval data from TROPOMI at finer time scales.

The Lorente et. al. correction, which handles seasonality with a static correction based on SWIR surface albedo, significantly improves upon the uncorrected data, but preserves the seasonal trend in the data, demonstrating larger, positive correlations in the winter months and cycling through the seasons.

*I'm also concerned about the use XCH4 as a predictor variable (which turns out to be the most important predictor) because it makes the correction to the variable depend on the variable itself – the authors expressed concern that using wind speed as predictor variable would propagate as an aliasing factor in inversions, but using XCH4 as a predictor variable seems worse in that regard.*

Reviewer 2 brought up similar concerns, so this is obviously an important point. The only reason to not include the XCH4 and XCH4_corrected variables is if there were a danger of erroneous retrievals, which would be flagged and removed anyway. The fact that these variables are always of high importance is reassuring that the built-in algorithm is mostly correct all of the time. If there were variably much larger differences in the TROPOMI retrieved values and the predicted values, then the importance of these variables would likely decrease substantially.

As for the issue of circular inputs: wind speeds are avoided because wind speed is an input in the flux divergence method of quantifying methane emissions. We did not want to use wind speed to predict the methane concentration, then use wind speed again to calculate methane emissions. However, the XCH4 and XCH4_corrected variables are only used in the calculation of methane concentration. The model-predicted value of methane concentration is used in the calculation of methane emissions. This is difficult to describe and understand in text, so please reference the following cartoon describing this, which has also been added to the SI:

[Figure]

Figure S5. Demonstration of circular input of wind speed variables. The top panel shows the input of wind speed and wind direction into the ML model, and again into the flux divergence method, potentially causing double counting problems. The same issue is not seen with the XCH4 and XCH4_corrected values because those are not incorporated into the flux divergence method - only the model-predicted XCH4 is an input.

*There is also a lot of chatty prose and repetition in this paper, some of which does not follow standard scientific practice of conciseness and focus. We don't need to hear about Mark Twain or Horace Greeley.*

These brief quotes are meant to illustrate important points within the manuscript.

1) The Greely quote is meant to showcase how important agriculture is in this region and how the landscape is grassland/high desert that has been converted to farmland. In our opinion, this is a potent point to make, demonstrating why this agriculturally-minded albedo correction is especially important in this region.

2) We provided the flow rates of the two rivers in this region and compared it to the Mississippi river to give a numerical idea of scale. However, we believe some readers might connect more with a verbal illustration. Twain's quote disparaging the waterways was meant to illustrate just how small and pitiful these rivers are in reality. If you're unfamiliar with the North and South Platte rivers, "river" is an extremely generous term, and the quotation was meant to give that impression.

We understand the reviewer's concern, so we have cut the Greeley quotation. However, since the Twain quotation directly addresses a reviewer's later concern, we have decided to keep this quotation along with removing repetitive text throughout the manuscript.

*Line 12, Abstract: not sure what 'many satellite products' means. UV/Vis retrievals indeed often assume fixed surface albedo, but that wouldn't apply to the TROPOMI methane retrieval which calculates its own surface albedo and would thus account for seasonality, unless I'm missing something.*

We have removed the ambiguous "many satellite products" and we have refined the language around the surface albedo correction. As this is the abstract, we cannot fully explain the deficiencies of current albedo corrections here, but we have also modified the language elsewhere to better explain the phenomenon we are attempting to solve.

In many regions, especially those with agriculture, surface reflectance depends on season, but corrections for this dependence are calculated from long-term averages.

Later in the results and discussion section we describe

This correction algorithm does account for some seasonality because the TROPOMI retrieved variables include the surface albedo SWIR which is used in the Lorente et al. algorithm to calculate a correction value. The seasonal correlation reappears after the Lorente et al. correction because this correction assumes that the relationship between surface albedo SWIR and correction value is static over time.

We believe this better describes what the reviewer expects; that the Lorente et al. correction is performed based on retrieved albedo values which will fluctuate seasonally. We add that the long-term nature of the correction calculation leaves out important seasonality in the relationship between the SWIR and the XCH4, which the reappearance of the correlation between these values in the winter data confirms, must change over time.

*Line 66: here and below, the description of machine learning seems pretentious to me. Here it's just being used as a non-parametric statistical fit.*

Yes, machine learning is just another method for curve fitting. In presenting this work at conferences we consistently asked the question "who here is familiar with machine learning?" and got very little positive response, so we wanted to include a very simple definition of machine learning to make the work more accessible to a wider audience. We recognize that our intent to simplify the description made our descriptions less accurate. We have overhauled the language in this section:

Machine learning is a branch of artificial intelligence where computers are trained to recognize patterns and make decisions based on data, somewhat like how humans learn from experience. Some machine learning models are considered a 'black box' because it can be difficult to understand how they make decisions. To address this, tools like SHapely Additive exPlanations (SHAP) help provide insights into how machine learning models arrive at their predictions. (Lundberg and Lee, 2017; Rudin, 2019). Neural networks, a type of machine learning model, are inspired by the way the human brain processes information. They consist of layers of 'neurons' that work together to identify patterns in data(Martín Abadi et al., 2015).

*Line 174: here and elsewhere, a driving motivation for the paper is to apply seasonality to the albedo correction from Lorente et al. 2021. I couldn't find a description of the Lorente correction but my understanding is that it is an improved polynomial spectral representation of the albedo co-retrieved with methane. If so it would have seasonality.*

We are not attempting to apply seasonality to the Lorente et al correction, we are developing a novel method that provides a finer time resolution seasonal correction. You are correct as to the representation of the Lorente et al. correction and its existing seasonality. However, as discussed above, this correction was constructed using long-term averaged data and our evidence shows that the relationship between surface albedo SWIR and the correction value should change over time. Figures 2, 3, and 7 in the manuscript all provide Lorente et al. data, and all show that it is affected by seasonal albedo effects.

The authors recognize that this was not made clear in the original submission and our poor choice of language contributed to the reviewers confusion here. We have significantly updated the language of the manuscript.

In many regions, especially those with agriculture, surface reflectance depends on season, but corrections for this dependence are calculated from long-term averages.

When applied to methane retrievals on a seasonal basis, we show that some residual albedo effects are still apparent and may thus bias seasonal data.

In this work we demonstrate that seasonal or monthly averaged methane retrievals continue to be biased by albedo effects after the implementation of these correction algorithms.

When the data are shown by season, this is no longer true - Pearson correlations with an absolute value greater than 0.1 indicate that there exists some correlation between the SWIR surface albedo and the albedo-corrected methane retrieval. This correction algorithm does account for some seasonality because the TROPOMI retrieved variables include the surface albedo SWIR which is used in the Lorente et al. algorithm to calculate a correction value. The seasonal correlation reappears after the Lorente et al. correction because this correction assumes that the relationship between surface albedo SWIR and correction value is static over time.

Regardless of the reason, the shifting albedo and seasonally variable albedo effect biases methane retrieval data from TROPOMI at finer time scales.

The Lorente et. al. correction, which handles seasonality with a static correction based on SWIR surface albedo, significantly improves upon the uncorrected data, but preserves the seasonal trend in the data, demonstrating larger, positive correlations in the winter months and cycling through the seasons.

*Line 294, Figure 5: it would be good to show the actual TROPOMI data before the correction. Data over rivers are bad and really should be removed from the dataset, but maybe they are not flagged as bad in the retrieval?*

The effect observed in Figure 5 is due to the agriculture surrounding the rivers as described in the manuscript. Spot checking a number of files for the area, no pixels were flagged as "water" and only rarely as "some_water" the criteria definition for which is "Pixel contains water (however small the fraction), i.e. at least one of the 15 × 15 arcsecond subpixels in the SDP dataset is classified as water" in the TROPOMI documentation. This means that the area of water inside each pixel is sufficiently small as to not bias the retrieval data. "River" is a very generous term when describing the North and South Platte, where the widest parts of the waterways are on the order of

150 and 50 meters respectively. These waterways are paltry and not navigable and do not trigger any quality flags in TROPOMI retrievals. We have strengthened the language here to make this point more clear.

As figure 5 is focused on correction magnitude instead of methane concentration, we feel it is more appropriate to include uncorrected data in figure 7 instead, reproduced below.

The North and South Platte rivers are extremely small (average discharges 1,355 and 175 cu ft/s respectively, Mississippi river is 593,000 cu ft/s) and are far less in extent than one satellite pixel, making the flagging or removing of this data due to water content, unnecessary.

[Figure]

---

## Author Comment (AC2)

*Reviewer comments are italicized*

    Author responses are indented and unitalicized

    Changes to the manuscript text are in blue

    The authors would like to thank the anonymous reviewers for their time and attention to this manuscript. Your comments and suggestions have been greatly appreciated and have significantly improved the paper.

**Response to anonymous referee 2**

*Major comments*

*1) Throughout the paper, the authors are muddled in their explanations and terminology about the seasonality of the albedo correction in their work and the previous works by Lorente et al. and Balasus et al.. As I understand it, in both of these previous works the corrections applied to the retrieved XCH4 do vary over time (as albedo and other variables change), it's just that the models used for calculating these corrections use all available data and do NOT vary with time. Bradley et al. are quite careless in their terminology throughout in describing this concept and it is misleading and confusing for the reader.*

    The authors thank the reviewer for their careful review of the manuscript. We now recognize the confusing terminology used in the manuscript and have worked to provide both an explanation for the time-variance of the albedo dependence as well as describe the difference in methods better.

    ...in many regions, especially those with agriculture, surface reflectance depends on the season. Existing corrections for this effect do not take into account a changing relationship between reflectance and the methane correction value over time.

    When applied to methane retrievals on a seasonal basis, we show that some residual albedo effects are still apparent and may thus bias seasonal data.

    In this work we demonstrate that seasonal or monthly averaged methane retrievals continue to be biased by albedo effects after the implementation of these correction algorithms. When the data are shown by season, this is no longer true - Pearson correlations with an absolute value greater than 0.1 indicate that there exists some correlation between the SWIR surface albedo and the albedo-corrected methane retrieval. This correction algorithm does account for some seasonality because the TROPOMI retrieved variables include the surface albedo SWIR which is used in the Lorente et al. algorithm to calculate a correction value. The seasonal correlation reappears after the Lorente et al. correction because this correction assumes that the relationship between surface albedo SWIR and correction value is static over time.

    Regardless of the reason, the shifting albedo and seasonally variable albedo effect biases methane retrieval data from TROPOMI at finer time scales.

    The Lorente et. al. correction, which handles seasonality with a static correction based on SWIR surface albedo, significantly improves upon the uncorrected data, but preserves the seasonal trend in the data, demonstrating larger, positive correlations in the winter months and cycling through the seasons.

*2) The above point raises important (linked) questions that Bradley et al. do not satisfactorily discuss in the manuscript:*

- *What is the physical justification for applying a monthly-varying bias correction model as opposed to a fixed model? I assume that it's ostensibly that the relationship between the satellite bias and the predictor variables changes with time, but why is this exactly?*

We do not know the exact reasons for the time dependence between satellite bias and predictor variables. We have a few plausible ideas that may be whole or part of an explanation: seasonal differences in sunlight angle, the effects of partial snow cover, lower boundary layer differences in the winter such as aerosol optical depths. The actual reason is likely a varying combination of these and other factors, which is why factors related to these were included in the ML model. The Lorente et. al. correction does not take into account these other factors, while the Balassus et al. correction does, but does not account for the possibility that the relationships between these factors may change on a seasonal timescale.

- *The monthly DTL models appear to assign different relative import to the input variables in the different seasons, but why is this the case, and what is the implication of this?*

The difference in relative importance of the variables is the reason why the seasonal correction is necessary. Before performing this analysis, we did not know if the importances would change between seasons, and if they did not, then more than likely the long term model developed by Balasus et al. would have been sufficient. The change in importance is related to our previous explanation that the relationships between the different seasonal factors change. For example: In the summer months, aerosol optical thickness may be more important because the hot days and cooler nights introduce a large change in the value, whereas during winter the values may be more static. And perhaps the higher overall particulate concentrations in winter produce a bias related to sunlight angle, which makes the solar zenith and relative azimuth angles more important.

- *Why exactly is the temporally-fixed model of Balasus et al. for example, not sufficient in this case (beyond the fact that it is trained on global rather than regional data)?*

As noted above, the differences in importance between predictor variables in different monthly models showcases why the temporally-fixed Balasus et al. model is insufficient. They incorporate more important predictor variables than Lorente et al. so their model performs better. Additionally, the incorporation of these other variables (and their temporal variability) allows for improved seasonal resolution. However, the static importances hinder this seasonal resolution because the relationships between the predictor variables are not static over time.

*As a non-expert in ML methods, I can't comment on the technical detail, but in other data-fitting schemes there is the concept of 'overfitting' one's model to the 'true' or observational data. Is there a similar possibility here, where applying the DTL models separately to each month's data might destroy the conceptual generality
of the model?*

The reviewer is correct in that there is a danger of overfitting, which we carefully considered and analyzed while training the model. The DTL method we employed is relatively safer in regards to overfitting than other techniques due to the layer freezing while retraining the monthly models. Additionally, we measured the difference between training and validation loss values as a function

of training epoch and presented these in fig. S3 in the supplemental information. These plots show consistent decreases in both the training and validation loss up to a plateau. Overfitting would be visualized in these plots as an increase in validation loss as epochs progress, which we do not observe.

The gap between the training and validation loss, though relatively small, does indicate we are dealing with the opposite problem: underfitting. Unfortunately, the solution to underfitting is introducing more data to the model, which we are in limited supply of due to the low amount of overlap between GOSAT and TROPOMI in this region. Underfitting in this case means that our model could still be improved by incorporating more data, but the output values of the model are valid for our purposes. We have added text to the manuscript to describe where readers can get information related to overfitting in our data

To monitor against overfitting, training and validation loss for the training period of each model were calculated and are presented in Figure S3.

And further explanation on how to interpret the plots has been added to the SI

To ensure the monthly models are not over- or under-fitting the data, training and validation loss plots were constructed for the training process for each monthly model. The number of epochs for these training processes are generally quite low, but the lack of loss value fluctuation later in the training process demonstrates that more training is unnecessary. Models that are underfit will show significant gaps between the training and validation loss, and one could argue that the models presented here show some signs of underfitting. However, the solution to underfitting is generally to incorporate more data into the model, and the co-located GOSAT and TROPOMI points used for training were already stretched to their limit. An overfit model will appear in these plots as a divergence, generally later in the training process, of the training and validation losses. These models do not demonstrate any patterns of overfitting, suggesting that the models were trained accurately and successfully.

*3) The authors here build on and compare to Balasus et al.'s global correction model and they have changed 3 major factors in comparison to that model – the regional bounds covered by the training data; the use of the DTL method for monthly models; and finally, the chosen subset of predictor variables. Therefore, it's impossible to say how much of the improvement is due to the specific regional training and DTL method, and how much is simply due to the fact that (e.g.) XCH4 and XCH4_corrected are now input variables (and other variables have been removed). Is it possible to include results from a version of your regional DTL models that keeps the same predictor variables as Belasus et al.?*

Balasus et al. did not provide a rationale for their selection of predictor variables, so we made our own choices and described why in the machine learning methods section. The differences between our selection and theirs is our inclusion of pressure-related variables and the XCH4 and XCH4_corrected variables (which we will discuss next) as well as the exclusion of wind speeds. Our exclusion of windspeeds was discussed in the manuscript as an attempt to avoid circular use of input variables as our model was designed for later use with the flux divergence method for methane quantification, which uses wind speeds as input. We included pressure-related variables because these are related to altitude, which is already an input variable, but could produce additional insights.

[Figure]

| | |
|---|---|
| —— This Work | —— Uncorrected |
| —— Lorente et. al. | —— Balasus et. al. |

We calculated a new model using only the Balasus et al. variables and the results are presented here. The point in March that exceeds the pearson value threshold could almost certainly be lowered to below the threshold if we spent significantly more time optimizing the model (3-4 weeks of time) as our original model contained points of this nature before significant optimization as well. Even with this one exceedance, it is clear that the model we built with the Balasus et el. variables is better than the original Balasus et al. model, which uses different model architecture and a much wider spatial extent, indicating that it is our methods and not just our choice of variables that improve upon their work. Due to this, we prefer to keep our methods as originally presented along with the explanations for the different choices we made.

*4) The inclusion of XCH4 and XCH4_corrected is not justified in this work and should be. The authors state that they stopped using wind speed variables to avoid circular use of inputs in potential top-down application of this data, but shouldn't the use of XCH4 itself should be avoided for a similar reason? Especially as XCH4_apriori, XCH4_corrected and XCH4 turn out to be 3 of the 5 most important predictor variables.*

The only reason to not include the XCH4 and XCH4_corrected variables is if there were a danger of erroneous retrievals, which would be flagged and removed anyway. The fact that these variables are always of high importance is reassuring that the built-in algorithm is mostly correct all of the time. If there were variably much larger differences in the TROPOMI retrieved values and the predicted values, then the importance of these variables would likely decrease substantially.

As for the issue of circular inputs: wind speeds are avoided because wind speed is an input in the flux divergence method of quantifying methane emissions. We did not want to use wind speed to predict the methane concentration, then use wind speed again to calculate methane emissions. However, the XCH4 and XCH4_corrected variables are only used in the calculation of

methane concentration. The model-predicted value of methane concentration is used in the calculation of methane emissions. This is difficult to describe and understand in text, so please reference the following cartoon describing this:

[Figure]

The cartoon has also been added to the SI with the following figure caption:

Figure S5. Demonstration of circular input of wind speed variables. The top panel shows the input of wind speed and wind direction into the ML model, and again into the flux divergence method, potentially causing double counting problems. The same issue is not seen with the XCH4 and XCH4_corrected values because those are not incorporated into the flux divergence method - only the model-predicted XCH4 is an input.

*Specific comments*
*Line 28: This Karakurt et al. reference is quite dated now – can you cite more up-to-date values?*

Agriculture is the largest contributor to global anthropogenic methane emissions (41.0%), followed by the energy sector (38.4%) (Global Methane Tracker 2023 – Analysis, 2024).

*Line 36: Quantify 'relatively large'.*
We have clarified the language here to show how large uncertainties can be and why

Due to the prevalence of super-emitters skewing averages, both bottom-up and top-down methods have large and difficult to quantify uncertainties (sometimes well over 100% (Riddick et al., 2024)), especially when diverse sources of methane overlap (Allen, 2016, 2014).

*Line 38: Note the assumptions and limitations of bottom-up methods also.*

Bottom-up inventories rely on accurate reporting of emissions and emissions factors from private companies, and are extremely sensitive to super-emitting events that make up a minority of events, but a majority of the emissions (Allen, 2014; Riddick et al., 2024).

*Lines 48-49: These statements about the limitations of current albedo corrections need a reference.*

The statements on the limitations of current albedo corrections are based on the work presented in this manuscript. We removed one of the statements and refined the language of the other to make it clear that it is in this work that it will be shown.

There have been several recent updates to the dataset to mitigate this albedo effect using TROPOMI retrieval data over areas without emissions, and also by comparison with proxy retrievals from GOSAT, which are unaffected by surface albedo (Balasus et al., 2023; Lorente et al., 2021). When applied to methane retrievals on a seasonal basis, we show that some residual albedo effects are still apparent and may thus bias seasonal data. This study attempts to develop a seasonal albedo correction for the area of the Denver-Julesburg (DJ) basin in Colorado to account for these effects.

*Line 59: Clarification needed here – "cropland seasonal albedo shifts are not currently compensated for in albedo corrections". However, as I understand it, the other correction methods should indeed account for the changing albedo (which is retrieved for each sounding from the satellite and included in their correction models) – it's the model for the magnitude of the correction which is consistent over time.*

The authors agree that there should be more clarification here. However, because of the same issue as in the reviewers previous comment - our evidence for this is presented in the manuscript - we will add only a minor clarification here, then extend the discussion later in the manuscript to clarify further.

Agriculture complicates the measurement and attribution of methane emissions data in two major ways: 1) cropland seasonal albedo shifts are under- compensated for in current albedo corrections,

And later in the results and discussion we have added:

This correction algorithm does account for some seasonality because the TROPOMI retrieved variables include the surface albedo SWIR which is used in the Lorente et al. algorithm to calculate a correction value. The seasonal correlation reappears after the Lorente et al. correction because this correction assumes that the relationship between surface albedo SWIR and correction value is static over time.

*Line 76: If you want to stop including the year for Lorente et al.'s and Balasus et al.'s work from this point, you should now clearly state that the shorthand that you're using thereafter refers to a particular paper (with year cited).*

Albedo corrections for the TROPOMI methane data have been described in the literature, with the most prominent being from Lorente et. al. which is now incorporated into the TROPOMI retrieval algorithm. Another effective albedo correction is from Balasus et. al. which also utilizes machine learning. Lorente et. al. used a B-spline interpolation of albedo dependence calculated over 2 years of data, while Balassus et al. trained a machine learning model on global data over a multi-year timescale, using the University of Leicester (UoL) Greenhouse Gases Observing Satellite (GOSAT) proxy retrievals as the target data (Balasus et al., 2023; Lorente et al., 2021). We will refer to these corrections often as simply Balasus et al. or Lorente et al. corrections.

*Line 76: What do you mean by 'well-formed'?*
By "well-formed" we meant "effective" so we made this change
Another effective albedo correction is from Balasus et. al. which also utilizes machine learning.

*Line 92: 'different numbers of averaging kernels' – Is this correct as stated? Is it clearer to say that the (number of/locations of) vertical retrieval pressure levels are different for the two satellites, and therefore the averaging kernels are different?*
The statement is correct as stated, the two satellites have different numbers of averaging kernels, but this does put the kernels in different places. We have clarified the language here for readers who may be less familiar with the concept of an averaging kernel.
The two satellites used in this study, TROPOMI and GOSAT, have different spatial resolutions both in a latitude-longitude grid, but also vertically with different numbers of vertical retrieval pressure levels, known as averaging kernels.

*Line 106: Define QA.*
QA is an internal value provided by the TROPOMI retrieval. Each data point is subject to quality checks having to do with snow, cloud cover, proximity to swath edge, etc. We have made the meaning clearer in the manuscript.
We used level 2 reprocessed and offline version 2.4 methane column data, XCH4, with internal TROPOMI-defined QA values of >= 0.5, indicating good-quality retrievals and better, including good-quality snow-covered scenes, and the Short Wave Infra-Red (SWIR) surface albedo as co-retrieved with XCH4.

*Line 107: SWIR is defined later in the text but should be defined here (or possibly earlier?).*
We used level 2 reprocessed and offline version 2.4 methane column data, XCH4, with internal TROPOMI-defined QA values of >= 0.5, indicating good-quality retrievals and better, including good-quality snow-covered scenes, and the Short Wave Infra-Red (SWIR) surface albedo as co-retrieved with XCH4.

*Line 112: Make clear here that this is not something that you do in this study, however, due to your implicit striping correction?*

A well-known artefact in methane retrievals from TROPOMI is striping caused by small differences between across-track pixels, which can be mitigated by performing a stripe correction (Liu et al., 2021). This work utilizes an inherent stripe correction instead of a separate explicit stripe correction.

*Line 119: Add more detail here about the co-location criteria so the reader does not have to refer to the Balasus paper.*

We have updated the text to reflect both the Balasus et al. criteria and our updated point co-location criteria to make it easier for readers to find.

GOSAT pairs was based on that of Balasus et. al. where pairs are calculated as pixel centers <5 km apart in space and <1 hr apart in time (Balasus et al., 2023). As our method requires a large amount of data and the region is much smaller we loosened the criteria to any pixel overlap in space and 2 hr apart in time.

*Line 121: This Kriesel reference needs more information for the reader to find it. Indeed the author himself explicitly asks that any citation includes the URL, and states a year. http://www.dkriesel.com/en/science/neural_networks. There are other locations where the authors should be more careful and explicit with their citations.*

We have gone back through our citations and made a number of corrections.

*Line 127: Are these changes really 'minor'? See earlier major comment.*

The reviewer is correct in that the XCH4 and XCH4 corrected values become important, so "minor" may not be the best term to use. Because we are only changing 4 or 5 out of 30 variables, this is the minor change we referred to. Instead we have opted for "a few" to indicate the low-in-number changes without commenting on the magnitude.

The variables were selected based on previous ML work on this topic with a few changes (Balasus et al., 2023).

*Line 128: To be clear here – what exactly is the 'corrected XCH4'? The Lorente et al. bias correction?*

The reviewer is correct and we updated the language to make this clearer.

We chose to incorporate the retrieved and corrected XCH4, which is corrected based on the onboard albedo correction from Lorente et al.

*Line 132: As a non-expert, I need more clarification of what the hidden layers/levels are.*

More explanation as to what hidden layers are has been added to the manuscript

A neural network can be described as "deep" if it has 3 or more "hidden layers" or levels in the network. Hidden layers are the strata of neurons which receive input from above and output to below and are "hidden" because the only layers the user interacts with are the top level input and the bottom level output while there may be hundreds or even thousands of layers sandwiched between.

*Line 134: More detail needed here. Trained and tested on 80% of the data? How much of this was training data and what was the other 20% used for? How was the 80% selected? Was it evenly*

*distributed across months – or do months potentially get different amounts of training data? Would that be an issue if so?*

Thank you for pointing out this confusion, as it was misstated in the manuscript. The models were trained on 80% of the data and the model validation was performed with the remaining 20%. The 80% of points were randomly sampled from the total ~17,000 points and the months are not all equally represented due to fewer good-quality retrievals in winter months. This unequal distribution will skew the annual model to better predict corrections in warmer months, but our technique of transfer learning with monthly datasets removes this bias in our final product. Though the colder months will have fewer points, training the monthly models on only those points serves to remove the bias introduced by the annual model. We have updated the language to reflect this.

A Deep Transfer Learning (DTL) method was used where an annual base model was trained on 80% of the total points, randomly sampled.. These same points were then separated by the month of their collection, and used to train 12 separate monthly models, starting from the annual base model; a schematic representing this training process is presented in Fig. 1. The remaining 20% of the data were used to calculate final fit statistics for each monthly model. There is an unequal distribution of points across the months, which introduces some seasonal bias to the annual model. This bias is subsequently removed when the monthly models are trained.

*Lines 192 – 197: I found this information a little hard to follow as structured here. Can you make clearer?*

The point of this section is to explain how we can compare each of the lines despite them using different datasets in the Pearson correlation. We agree that text-format makes this difficult to describe and understand. We have opted to remove most of the explanatory text in the manuscript in favor of a much more concise table which has been added to the supplemental information. The explanation is not necessary to understand the figure, but is important for certain readers who may want to construct a similar figure to understand exactly how it was done.

To evaluate against the other models, Pearson correlations were calculated and presented in Fig 3a where different constructions of Pearson values have been unified according to Tab. S2.

| | Axis | Explanation |
|---|---|---|
| All Lines | x | Surface Albedo SWIR |
| Uncorrected | y | Proxy retrieval GOSAT / TROPOMI XCH4 |
| Lorente et al. | y | Proxy retrieval GOSAT / TROPOMI XCH4_corrected |
| Harvard | y | Sensitivity adjusted GOSAT / Balasus et al. corrected XCH4 |
| This Work | y | Sensitivity adjusted GOSAT / This work corrected XCH4 |

This table outlines the construction of the axes in Fig. 3a. The main difference is whether the raw proxy retrieval data were used or whether the sensitivity-adjusted proxy retrieval data were used. This determination was based on the construction of the correction algorithm - the raw XCH4 measurement and the Lorente et al. corrected measurement were both constructed using the raw proxy retrieval GOSAT data, so that is what they were compared against for the Pearson correlation. The Harvard dataset and this work were both constructed using the sensitivity-adjusted proxy retrieval GOSAT data, so these Pearson correlations were calculated using these alternate values.

*Line 211: 'not designed to handle seasonality' – rephrase.*

Amended

The Lorente et. al. correction, which handles seasonality with a static correction based on SWIR surface albedo, significantly improves upon the uncorrected data, but preserves the seasonal trend in the data, demonstrating larger, positive correlations in the winter months and cycling through the seasons.

*Around line 265 – what does it mean that the relative importance of predictor variables in your work appears quite different to those of Belasus et al.? (e.g. Aerosols/Surface Albedo SWIR are very important for them).*

We have added an explanation to the manuscript describing why there are differences in importances between our work and Balasus et al. It is likely due to the different extents and the ranges of values we have access to in just the Denver-Julesburg basin vs. the global extent that Balasus et al. used.

The importances of variables here differ from the importances determined in Balasus et al. likely due to extent. This work's much smaller area focused on the Denver-Julesberg basin has a very limited range of surface albedo SWIR values, whereas the Balasus et al. global extent sees a range of 0.01-0.6 in some regions. The much smaller range of SWIR surface albedo here likely contributes to the lower overall importance. The extent likely also affects the importance of aerosol-related variables, which Balasus et al. also found to be significantly more important – our extent focused on the oil and gas basin with significant agricultural influence, which are two important sources of aerosols, but our proximity to sources may limit the range of aerosol-related values, making this term less important here than on a global extent as well.

*Figure 5: Discuss – why are the urban corrections so variable? Should they be relatively constant?*

The urban corrections are variable along with the other land uses because of the changing relationship between surface albedo and other variables with the correction value in time. Urban SWIR albedos also change in time, though not on the same timescale as croplands, but this correction is not just based on seasonal changes in albedo, but also the seasonal changes in the relationship between albedo and the correction value. We have added a short discussion of this to the manuscript as well.

That the urban points also follow the general seasonal trend is important and indicates that a driving factor in the seasonal albedo change is the relationship between surface albedo SWIR and other variables with the correction value and how that relationship changes seasonally.

*Line 325: To be clear – is this testing the significance of the differences between derived corrections over the different land-types?*

Yes, that is correct. The significance of this plot is that the darker squares are located more on the left and right edges of the plot, indicating that in colder months there is less statistical difference between the land use types than during the summer. Except for urban land use types which are different throughout the year.

*Line 336: I still find this reasoning confusing – I apologise if I'm misunderstanding. Other corrections do take account of the changing albedo of the crops, which is retrieved by TROPOMI, right? So the reason that they're not capturing the seasonal variability is their model? But why is the relationship between bias and albedo changing with time?*

You are not misunderstanding and your commentary has been very helpful on this point! This point has been addressed above with the major comments.

*Figure 7: I think I'm right in saying that figure captions in this journal should include a description of the figure only and not 'discussion'-type text such as this one comparing results in the different panels.*

Figure 7. Result of the methane albedo correction devised in this work. The Lorente et. el. correction in the summer (a) is compared against the correction devised by this work (c) and the average surface albedo SWIR retrieval map for this time period (b) This is repeated for the winter months on the right with the Lorente et al. correction (d), this work's correction (f), and the winter average surface albedo SWIR (e)

*Line 346: 'appears to have' – you should be able to quantify this.*

The corrected dataset devised here has an average mixing ratio 6.9 ppb smaller than the Lorente et al. corrected data in the summer and 0.4 ppb smaller in winter, appearing slightly darker in color than the Lorente corrected data. This is likely due to the correction's dependence on the UoL GOSAT proxy retrieval target data, which on a global average measures 9.2 ppb less XCH4 than TROPOMI (Balasus et al., 2023). Notably, the Denver metropolitan area has lower average methane concentrations in our model output data than in the original TROPOMI Lorente et al. corrected data (6.4 ppb less in summer and 4.9 ppb less in winter).

*Line 347: 'due to the correction's dependence on the UoL GOSAT … data' – you should confirm that the Lorente data is based on GOSAT? I was under the impression that Lorente et al.'s correction was not based on GOSAT.*

The reviewer is correct that Lorente et al.'s correction is not based on GOSAT, this phrase was meant to refer to the correction we formed in this work. We have changed the language to better reflect our meaning.

This reduction is likely due to the new correction algorithm's dependence on the UoL GOSAT proxy retrieval target data, which on a global average measures 9.2 ppb less XCH4 than TROPOMI (Balasus et al., 2023).

*Line 350: 'appear visually smoother' this is vague and unscientific.*

The phrase has been removed

*Lines 351 – 357: I'm slightly unclear as to what you are saying with this statement regarding the lack of necessary GOSAT data. Can you clarify further? Is it that you are not able to produce a map of correlations?*

The correlations presented earlier in the paper are correlations between a ratio of GOSAT/TROPOMI and the SWIR surface albedo. In order to generate similar correlations here, we would have to have GOSAT data covering the same spatial extent with similar point density. These

maps were calculated over three months each and GOSAT data is sparse and infrequent, so this is not possible. The GOSAT data are required because there are natural correlations between surface albedo and methane concentration, so calculating GOSAT/TROPOMI removes the bias caused by these natural (real) correlations.

Figure 7 cannot be evaluated as before with a Pearson correlation because the correlation requires GOSAT / TROPOMI data to be used to account for natural correlation between surface albedo SWIR and XCH4. There is not sufficient GOSAT data over this extent and time period to calculate such a Pearson correlation.

*Technical corrections*
*Line 35: The (number/density/range?) of atmospheric measurements has expanded…*

Atmospheric measurements have expanded in both number and choice of platform with the advances of satellite monitoring systems (de Gouw et al., 2020; Jacob et al., 2016).

*Line 43 and elsewhere : Include hyphen: "Sentinel-5 Precursor"*
Amended

*Line 47: "currently-used"*
Sentence removed for other reasons

*Line 51: top 10 U.S. states*
Colorado ranks in the top 10 U.S. states in total energy production (U.S. EIA, 2020).

*Line 61: change to: (CAFOs)*
Amended

*Line 64 and elsewhere: be consistent in spelling of collocation/co-location throughout.*
Amended

*Line 275: "Model-predicted"*
Amended

*Line 277: Surely red, not brown? Additionally, it would be good if Figure S1 used the
same red-blue colourbar if possible. The green colours (ranging from around +5 to -15
ppb) are hard to distinguish from each other.*

In a comedic turn of events, both the main author and the main reader of this manuscript are red-green colorblind. This is one of the rare instances where us colorblind folk should have made sure our manuscript was accessible to normal color-vision readers.

---

## Author Comment (AC3)

*Reviewer comments are italicized*

      Author responses are indented and unitalicized

      Changes to the manuscript text are in blue

The authors would like to thank the anonymous reviewers for their time and attention to this manuscript. Your comments and suggestions have been greatly appreciated and have significantly improved the paper.

**Response to anonymous referee 3**

*G1: I find that the biases, which the authors are trying to correct for, appear mostly linear (see Figure 2). I am not convinced that using a machine learning correction is necessary and I fear that the introduction of such highly non-linear models will lead to overfitting. How have you ensured that you are fitting signal and not noise?*

      A Machine learning model does not try to impose non-linearity upon the fit, so if a linear fit is most appropriate according to the model training, then the resulting model will appear more linear. Using the machine learning model could be computational overkill, but the results would not be negatively affected.

      Overfitting can be monitored for in the training process of the ML models and we did so to ensure we were not overfitting. The results of which are available in the Supplementary Information figure S3, which describe the training and validation loss throughout the training process for each monthly model. Overfitting would be identified in these plots as a significant increase in validation loss while training loss remains low. The consistent gap between the training and validation loss at the end of the training period actually indicates we are underfitting with our models. Unfortunately, the solution to underfitting is introducing more data to the model, which we are in limited supply of due to the low amount of overlap between GOSAT and TROPOMI in this region. Underfitting in this case means that our model could still be improved by incorporating more data, but the output values of the model are valid for our purposes.

      To clarify this for readers, we have added the following to the methods section:

      "To monitor against overfitting, training and validation loss for the training period of each model were calculated and are presented in **Figure S3**."

      And we have added more text to explain the SI figure further to ensure readers understand there has been no overfitting

      "To ensure the monthly models are not over- or under-fitting the data, training and validation loss plots were constructed for the training process for each monthly model. The number of epochs for these training processes are generally quite low, but the lack of loss value fluctuation later in the training process demonstrates that more training is unnecessary. Models that are underfit will show significant gaps between the training and validation loss, and one could argue that the models presented here show some signs of underfitting. However, the solution to underfitting is generally to incorporate more data into the model, and the co-located GOSAt and TROPOMI points used for training were already stretched to their limit. An overfit model will appear in these plots as a divergence, generally later in the training process, of the training and validation losses. These models do not demonstrate any patterns of overfitting, suggesting that the models were trained accurately and successfully."

*G2: I am very surprised to see such strong dependence of the XCH4 bias correction on land cover type, which is much more typically observed (and intuitively understood) in trace gas retrievals from coarse-spectral-resolution sensors like AVIRIS-NG. Please comment in the text why such hyper-local corrections might be physically plausible.*

The XCH4 bias correction is strongly dependent on very specific land cover types. The referee will notice that "Non-Agricultural" land cover types include a multitude of cover types and the variability within this group is quite small. Where we observe strong dependence is between water-intensive and drought-resistant crops and we believe this is due to the soil or vegetation water content and its effect on the measurement light frequencies. This is already described in the manuscript:

"...water sources like these two eventual tributaries to the Missouri River dictate where larger water-intensive agricultural operations exist. As such, larger densities of water-intensive crop farms are co-located with these rivers, bringing their albedo-influencing crops and plant-life, and thus requiring an albedo correction which is not necessarily reflected in magnitude by the surrounding scrubland. It has been shown that water intensive crops, like corn, sugarbeets, and alfalfa; and drought resistant crops, like winter wheat, millet, and dry beans; reflect SWIR light differently, allowing for identification of crops from space with the SWIR reflectance variable along with other variables (Chen et al., 2005). This effect is possibly due to water content or leaf size of the vegetable matter."

*G3: The Pearson correlation coefficient, which has been chosen as a metric for the performance of different albedo corrections here, is hard to interpret (correlation between albedo and Delta_XCH4 (TROPOMI-GOSAT) is almost the same for the uncorrected data and the proposed correction in the month of December). Please explain your reasoning in more detail and provide additional metrics to measure how your new correction performs in comparison to the existing corrections by both Lorente et al. and Balasus et al..*

Theoretically there should be no correlation between the surface albedo and the methane mixing ratio. The fact that we observe a correlation means that the instruments and measurement techniques are introducing a bias, such as in the uncorrected, Lorente et al. and Balasus et al. lines in Figure 3. Our correction remains between -0.1 and 0.1 Pearson correlation value for all months, indicating that our correction is the least biased with respect to surface albedo. The uncorrected data being similar to our correction in December is not an issue, there is simply less albedo bias in the winter months, we believe this is related to agricultural land-use in the area not affecting winter months as much as this is not a growing season at this latitude.

We prefer not to devise a new metric, as this is the one used in Lorente et al. 2021 and is the most relevant metric to the topic of the manuscript. We have added more clarifying text to the manuscript to make interpretation of the Pearson correlation more clear.

"To evaluate against the other models, Pearson correlations were calculated and presented in Fig 3a where different constructions of Pearson values have been unified according to Tab. S2, where the Pearson correlations have been calculated the same ways as in Fig. 2 with the correlation between GOSAT/TROPOMI and surface albedo. To reiterate, a Pearson correlation of 0 is

the preferred value, as the difference between the two data sets does not depend on surface albedo. The surface albedo is the SWIR albedo as retrieved by TROPOMI."

*M1: Line 15-17: the 5-6 ppb correction occurs only at some times during the year. Be more specific here and in the text, for instance in Line 291-293.*

      Amended, the second instance was removed entirely.

      "requiring a correction of 5-6 ppb larger than areas covered in water-intensive crops in the summer."

*M2: Line 44-46: Rephrase this sentence to make sure that readers understand that a) the performance of proxy retrievals is not generally unaffected by albedo and b) GOSAT XCH4 can also exhibit some (weak) albedo bias, but less than TROPOMI XCH4 due to a number of reasons (imaging grating vs FTS, spectral resolution and band, etc.).*

      Amended, this is also repeated in line 116 of the original manuscript

      "There have been several recent updates to the dataset to mitigate this albedo effect using TROPOMI retrieval data over areas without emissions, and also by comparison with proxy retrievals from GOSAT, which are much less affected by surface albedo"

      "This dataset has been used extensively before as a measurement that is less affected by changing surface albedo"

*M3: Line 85-86: "Satellite methods for...are going to be biased..." -> "Satellite methods for...can be biased..."*

      Amended

*M4: Line 86-88: Rephrase, because CH4 emissions from oil and gas operations can be strongly time-dependent.*

      We prefer to keep this sentence as is, it is not uncommon to assume that emissions from O&G operations do not fluctuate considerably with the seasons, see:

      Wilson, C.; Chipperfield, M. P.; Gloor, M.; Chevallier, F. Development of a Variational Flux Inversion System (INVICAT v1.0) Using the TOMCAT Chemical Transport Model. *Geosci. Model Dev.* **2014**, *7* (5), 2485–2500. https://doi.org/10.5194/gmd-7-2485-2014.

      Shen, L.; Gautam, R.; Omara, M.; Zavala-Araiza, D.; Maasakkers, J. D.; Scarpelli, T. R.; Lorente, A.; Lyon, D.; Sheng, J.; Varon, D. J.; Nesser, H.; Qu, Z.; Lu, X.; Sulprizio, M. P.; Hamburg, S. P.; Jacob, D. J. Satellite Quantification of Oil and Natural Gas Methane Emissions in the US and Canada Including Contributions from Individual Basins. *Atmospheric Chemistry and Physics* **2022**, *22* (17), 11203–11215. https://doi.org/10.5194/acp-22-11203-2022.

*M5: Line 98-100: move this sentence to section 2.2 ?*

      Amended

*M6: Figs. 2,3/ Lines 175-181: Fig. 2 indicates that a small bias exists only in winter (|R| > 0.1). So based on your threshold in Pearson's correlation coefficient, no correction would be needed in*

*summer? This appears to be supported by Fig. 3 which seems to indicate that the Lorente et al. correction works well in 6 out of 12 months.*

The reviewer is correct, the Lorente et al. correction falls within the ideal Pearson value range for about half the year. This speaks further to the "seasonality" required for a more accurate correction. Indeed, the Balasus et al. correction falls within this range much of the time that the Lorente correction does not, so someone could put together a combined Balasus-Lorente correction and it may be mostly correct most of the time.

*M7: Line 198-201: Please provide more reasoning with respect to the choice of the threshold in |R| and possibly replace the reference Kuckartz et al. 2013, since this German reference may be a challenging resource for many readers (if you don't replace it, double-check the page number).*

The choice of Pearson R value is largely arbitrary, but was selected with the idea that essentially nobody would argue that a Pearson R value <0.1 was anything other than negligible. We have altered the language in the manuscript to reflect this point and have updated our references to more accessible and appropriate sources.

"When all data are used (Fig 2a) the Pearson correlation is indeed calculated to be low, i.e. below a threshold of 0.1, which we chose here as a target value for minimal correlation between SWIR surface albedo and the albedo corrected methane retrieval. Though the significance of Pearson coefficients is up to interpretation, most would agree that a value of <0.1 is negligible (Akoglu, 2018; Schober et al., 2018)."

Akoglu, H.: User's guide to correlation coefficients, Turk J Emerg Med, 18, 91–93, https://doi.org/10.1016/j.tjem.2018.08.001, 2018.

Schober, P., Boer, C., and Schwarte, L. A.: Correlation Coefficients: Appropriate Use and Interpretation, Anesthesia & Analgesia, 126, 1763, https://doi.org/10.1213/ANE.0000000000002864, 2018.

*M8: Fig. 4: Can you comment why albedo in the NIR appears to be more important than the albedo in the SWIR? This appears counter intuitive.*

We cannot say with 100% certainty why the NIR albedo appears to be more important than the SWIR albedo due to the translucent-box nature of the machine learning algorithm. As described in the manuscript, we define our correction as a SWIR albedo correction due to our choice of correction validation. Over the course of this work we trained dozens of sets of models using various hyperparameters and other changes, but the models we chose to use were the ones where the correlation between surface albedo SWIR and GOSAT/TROPOMI was minimized. This does not mean that the SWIR albedo is necessarily the most or even an important factor in the actual model correction. The higher importance of the NIR albedo suggests that the NIR albedo variable has a larger impact on the final output value than the SWIR albedo. While this does not make intuitive sense knowing that the SWIR albedo is more likely to affect the actual measurement of the XCH4, the model does not have this knowledge to bias its decision making. We have clarified this further in the manuscript.

"Figure 4 shows that other features may be more important than the surface albedo SWIR in the actual model calculation. "importance" in a ML model is the magnitude of effect that variable has on the final output value of the model. The variables that appear higher on the y-axis than "surface albedo SWIR" tended to be more important and should be analyzed as well. Some of these

variables have clear reasonings as to why they are more important: XCH4 apriori, XCH4 corrected, and XCH4 are all the measurements of methane mixing ratio that were either priors for the TROPOMI measurement (XCH4 apriori) or direct measurements of the methane mixing ratio by TROPOMI (XCH4 and XCH4 corrected). XCH4 and "XCH4 corrected" directly measure methane mixing ratios via TROPOMI, serving as primary data sources for our predictive models. The reasoning for other important variables is not so clear: "surface albedo SWIR precision" and "chi square SWIR". The precision of the surface albedo SWIR measurement being important was not expected, but may be the result of a well-trained model successfully making the association between the SWIR albedo measurement and its precision. A less precise measurement would be less heavily relied upon for the model's predictions, so the importance may come from the association between the precision measurement and how much a particular measurement affected the model during training."

Technical comments:
*T1: Line 55: "Kansas. (Petron et al…)." -> "Kansas (Petron et al …)."*
 Amended
*T2 : Line 61 : (CAFO)s -> (CAFOs)*
 Amended
*T3 : Line 64 : Collocation -> Colocation*
 Amended
*T4: Line 119 : Balasus et. al. (Balasus et al., 2023) -> Balasus et al. (2023)*
 Amended